# Coded Distributed Computing for Inverse Problems

**Yaoqing Yang, Pulkit Grover and Soummya Kar**
Carnegie Mellon University
{yyaoqing, pgrover, soummyak}@andrew.cmu.edu

## Abstract

Computationally intensive distributed and parallel computing is often bottlenecked by a small set of slow workers known as stragglers. In this paper, we utilize the emerging idea of "coded computation" to design a novel error-correcting-code inspired technique for solving linear inverse problems under specific iterative methods in a parallelized implementation affected by stragglers. Example machine-learning applications include inverse problems such as personalized PageRank and sampling on graphs. We provably show that our coded-computation technique can reduce the mean-squared error under a computational deadline constraint. In fact, the ratio of mean-squared error of replication-based and coded techniques diverges to infinity as the deadline increases. Our experiments for personalized PageRank performed on real systems and real social networks show that this ratio can be as large as $10^4$. Further, unlike coded-computation techniques proposed thus far, our strategy combines outputs of all workers, including the stragglers, to produce more accurate estimates at the computational deadline. This also ensures that the accuracy degrades "gracefully" in the event that the number of stragglers is large.

## 1   Introduction

The speed of distributed computing is often affected by a few slow workers known as the "stragglers" [1–4]. This issue is often addressed by replicating tasks across workers and using this redundancy to ignore some of the stragglers. Recently, methods from error-correcting codes (ECC) have been used for speeding up distributed computing [5–15], which build on classical works on algorithm-based fault-tolerance [16]. The key idea is to treat stragglers as "erasures" and use ECC to retrieve the result after a subset of fast workers have finished. In some cases, (e.g. [6, 8] for matrix multiplications), techniques that utilize ECC achieve scaling-sense speedups in average computation time compared to replication. In this work, we propose a novel coding-inspired technique to deal with stragglers in distributed computing of linear inverse problems using iterative solvers [17].

Existing techniques that use coding to deal with stragglers treat straggling workers as "erasures", that is, they ignore computation results of the stragglers. In contrast, when using iterative methods for linear inverse problems, even if the computation result at a straggler has not converged, the proposed algorithm does not ignore the result, but instead combines it (with appropriate weights) with results from other workers. This is in part because the results of iterative methods often converge gradually to the true solutions. We use a small example shown in Fig. 1 to illustrate this idea. Suppose we want to solve two linear inverse problems with solutions $\mathbf{x}_1^*$ and $\mathbf{x}_2^*$. We "encode the computation" by adding an extra linear inverse problem with solution $\mathbf{x}_1^* + \mathbf{x}_2^*$ (see Section 3), and distribute these three problems to three workers. Using this method, the solutions $\mathbf{x}_1^*$ and $\mathbf{x}_2^*$ can be obtained from the results of any combination of two fast workers that first return their solutions.

But what if we have a computational deadline, $T_{dl}$, by which *only one* worker converges? The natural extension of existing strategies (e.g., [6]) will declare a failure because it needs at least two workers to respond. However, our strategy does not require convergence: even intermediate results can be utilized to estimate solutions. In other words, our strategy degrades gracefully as the number of stragglers increases, or as the deadline is pulled earlier. Indeed, we show that it is suboptimal to ignore stragglers as erasures, and design strategies that treat the difference from the optimal solution

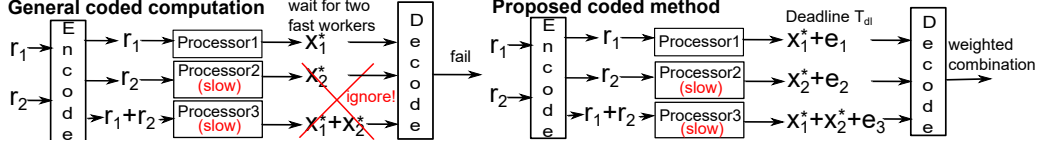

Figure 1: A comparison between the existing scheme in [6] and the proposed algorithm.

as "soft" additive noise (see Section 3). We use an algorithm that is similar to weighted least-squares for decoding, giving each worker a weight based on its proximity to convergence. In this way, we can expect to fully utilize the computation results from all workers and obtain better speedup.

Theoretically, we show that for a specified deadline time $T_{\mathrm{dl}}$, under certain conditions on worker speed distributions, the coded linear inverse solver using structured codes has smaller mean squared error than the replication-based linear solver (Theorem 4.4). In fact, under more relaxed conditions on worker speed distributions, when the computation time $T_{\mathrm{dl}}$ increases, the ratio of the mean-squared error (MSE) of replication-based and coded linear solvers can get arbitrarily large (Theorem 4.5)! For validation of our theory, we performed experiments to compare coded and replication-based computation for a graph mining problem, namely personalized PageRank [18] using the classical power-iteration method [19]. We conduct experiments on the Twitter and Google Plus social networks under a deadline on computation time using a given number of workers on a real computation cluster (Section 6). We observe that the MSE of coded PageRank is smaller than that of replication by a factor of $10^4$ at $T_{dl} = 2$ seconds. From an intuitive perspective, the advantage of coding over replication is that coding utilizes the diversity of all heterogeneous workers, whereas replication cannot (see section 7 for details). To compare with existing coded technique in [6], we adapt it to inverse problems by inverting only the partial results from the fast workers. However, from our experiments, if only the results from the fast workers are used, the error amplifies due to inverting an ill-conditioned submatrix during decoding (Section 6). This ill-conditioning issue of real-number erasure codes has also been recognized in a recent communication problem [20]. In contrast, our novel way of combining all the partial results including those from the stragglers helps bypass the difficulty of inverting an ill-conditioned matrix.

The focus of this work is on utilizing computations to deliver the minimal MSE in solving linear inverse problems. Our algorithm does not reduce the communication cost. However, because each worker performs sophisticated iterative computations in our problem, such as the power-iteration computations, the time required for computation dominates that of communication (Section 5.2). This is unlike some recent works (e.g.[21–24]) where communication costs are observed to dominate because the per-processor computation is smaller.

Finally, we summarize our main contributions in this paper:

- We propose a coded computing algorithm for multiple instances of a linear inverse problem;
- We theoretically analyze the mean-squared error of coded, uncoded and replication-based iterative linear solvers under a deadline constraint, and show scaling sense advantage of coded solvers in theory and orders of magnitude smaller error in data experiments.
- This is the first work that treats stragglers as soft errors instead of erasures, which leads to graceful degradation in the event that the number of stragglers is large.

## 2 System Model and Problem Formulation

### 2.1 Preliminaries on Solving Linear Systems using Iterative Methods

Consider the problem of solving $k$ inverse problems with the same linear transform matrix $\mathbf{M}$ and different inputs $\mathbf{r}_i$: $\mathbf{M}\mathbf{x}_i = \mathbf{r}_i, i = 1, 2, \ldots k$. When $\mathbf{M}$ is a square matrix, the closed-form solution is $\mathbf{x}_i = \mathbf{M}^{-1}\mathbf{r}_i$. When $\mathbf{M}$ is a non-square matrix, the regularized least-square solution is $\mathbf{x}_i = (\mathbf{M}^\top\mathbf{M} + \lambda\mathbf{I})^{-1}\mathbf{M}^\top\mathbf{r}_i, i = 1, 2, \ldots k$, with an appropriate regularization parameter $\lambda$. Since matrix inversion is hard, iterative methods are often used. We now look at two ordinary iterative methods, namely the Jacobian method [17] and the gradient descent method. For a square matrix $\mathbf{M} = \mathbf{D} + \mathbf{L}$, where $\mathbf{D}$ is diagonal, the Jacobian iteration is written as $\mathbf{x}_i^{(l+1)} = \mathbf{D}^{-1}(\mathbf{r}_i - \mathbf{L}\mathbf{x}_i^{(l)})$. Under certain conditions of $\mathbf{D}$ and $\mathbf{L}$ ([17, p.115]), the computation result converges to the true solution. One example is the PageRank algorithm discussed in Section 2.2. For the $\ell_2$-minimization problem with a non-square $\mathbf{M}$, the gradient descent method has the form $\mathbf{x}_i^{(l+1)} = ((1-\lambda)\mathbf{I} - \epsilon\mathbf{M}^\top\mathbf{M})\mathbf{x}_i^{(l)} + \epsilon\mathbf{M}^\top\mathbf{r}_i,$

where $\epsilon$ is an appropriate step-size. We can see that both the Jacobian iteration and the gradient descent iteration mentioned above have the form

$$\mathbf{x}_i^{(l+1)} = \mathbf{B}\mathbf{x}_i^{(l)} + \mathbf{K}\mathbf{r}_i, i = 1, 2, \ldots k, \tag{1}$$

for two appropriate matrices $\mathbf{B}$ and $\mathbf{K}$, which solves the following equation with true solution $\mathbf{x}_i^*$:

$$\mathbf{x}_i^* = \mathbf{B}\mathbf{x}_i^* + \mathbf{K}\mathbf{r}_i, i = 1, 2, \ldots k. \tag{2}$$

Therefore, subtracting (2) from (1), we have that the computation error $\mathbf{e}_i^{(l)} = \mathbf{x}_i^{(l)} - \mathbf{x}_i^*$ satisfies

$$\mathbf{e}_i^{(l+1)} = \mathbf{B}\mathbf{e}_i^{(l)}. \tag{3}$$

For the iterative method to converge, we always assume the spectral radius $\rho(\mathbf{B}) < 1$ (see [17, p.115]). We will study iterative methods that have the form (1) throughout this paper.

## 2.2 Motivating Applications of Linear Inverse Problems

Our coded computation technique requires solving multiple inverse problems with the same linear transform matrix $\mathbf{M}$. One such problem is personalized PageRank. For a directed graph, the PageRank algorithm [19] aims to measure the nodes' importance by solving the linear problem $\mathbf{x} = \frac{d}{N}\mathbf{1}_N + (1-d)\mathbf{A}\mathbf{x}$, where $d = 0.15$ is called the "teleport" probability, $N$ is the number of nodes and $\mathbf{A}$ is the column-normalized adjacency matrix. The personalized PageRank problem [18] considers a more general equation $\mathbf{x} = d\mathbf{r} + (1-d)\mathbf{A}\mathbf{x}$, for any possible vector $\mathbf{r} \in \mathbb{R}^N$ that satisfies $\mathbf{1}^\top \mathbf{r} = 1$. Compared to PageRank [19], personalized PageRank [18] incorporates $\mathbf{r}$ as the preference of different users or topics. A classical method to solve PageRank is power-iteration, which iterates the computation $\mathbf{x}^{(l+1)} = d\mathbf{r} + (1-d)\mathbf{A}\mathbf{x}^{(l)}$ until convergence. This iterative method is the same as (1), which is essentially the Jacobian method mentioned above. Another example application is the sampling and recovery problem in the emerging field of graph signal processing [25, 26] as a non-square system, which is discussed in Supplementary section 8.1.

## 2.3 Problem Formulation: Distributed Computing and the Straggler Effect

Consider solving $k$ linear inverse problems $\mathbf{M}\mathbf{x}_i = \mathbf{r}_i, i = 1, 2, \ldots k$ in $n > k$ workers using the iterative method (1), where each worker solves one inverse problem. Due to the straggler effect, the computation at different workers can have different speeds. The goal is to obtain minimal MSE in solving linear inverse problems before a deadline time $T_{\text{dl}}$. Suppose after $T_{\text{dl}}$, the $i$-th worker has completed $l_i$ iterations in (1). Then, from (3), the residual error at the $i$-th worker is

$$\mathbf{e}_i^{(l_i)} = \mathbf{B}^{l_i}\mathbf{e}_i^{(0)}. \tag{4}$$

For our theoretical results, we sometimes need the following assumption.
*Assumption* 1. We assume that the optimal solutions $\mathbf{x}_i^*, i = 1, 2, \ldots k$, are i.i.d.

Denote by $\boldsymbol{\mu}_E$ and $\mathbf{C}_E$ respectively the mean and the covariance of each $\mathbf{x}_i^*$. Note that Assumption 1 is equivalent to the assumption that the inputs $\mathbf{r}_i, i = 1, 2, \ldots k$ are i.i.d., because $\mathbf{r}_i$ and $\mathbf{x}_i^*$ are related by the linear equation (2). For the personalized PageRank problem discussed above, this assumption is reasonable because queries from different users or topics are unrelated. Assume we have estimated the mean $\boldsymbol{\mu}_E$ beforehand and we start with the initial estimate $\mathbf{x}_i^{(0)} = \boldsymbol{\mu}_E$. Then, $\mathbf{e}_i^{(0)} = \mathbf{x}_i^{(0)} - \mathbf{x}_i^*$ has mean $\mathbf{0}_N$ and covariance $\mathbf{C}_E$. We also try to extend our results for the case when $\mathbf{x}_i^*$'s (or equivalently, $\mathbf{r}_i$'s) are correlated. Since the extension is rather long and may hinder the understanding of the main paper, we provide it in supplementary section 8.2 and section 8.5.

### 2.4 Preliminaries on Error Correcting Codes

We will use "encode" and "decode" to denote preprocessing and post-processing before and after parallel computation. In this paper, the encoder multiplies the inputs to the parallel workers with a "generator matrix" $\mathbf{G}$ and the decoder multiplies the outputs of the workers with a "decoding matrix" $\mathbf{L}$ (see Algorithm 1). We call a code an $(n, k)$ code if the generator matrix has size $k \times n$. We often use generator matrices $\mathbf{G}$ with orthonormal rows, which means $\mathbf{G}_{k \times n}\mathbf{G}_{n \times k}^\top = \mathbf{I}_k$. An example of such a matrix is the submatrix formed by any $k$ rows of an $n \times n$ orthonormal matrix (e.g., a Fourier matrix). Under this assumption, $\mathbf{G}_{k \times n}$ can be augmented to form an $n \times n$ orthonormal matrix using another matrix $\mathbf{H}_{(n-k) \times n}$, i.e. the square matrix $\mathbf{F}_{n \times n} = \begin{bmatrix} \mathbf{G}_{k \times n} \\ \mathbf{H}_{(n-k) \times n} \end{bmatrix}$ satisfies $\mathbf{F}^\top \mathbf{F} = I_n$.

# 3 Coded Distributed Computing of Linear Inverse Problems

The proposed coded linear inverse algorithm (Algorithm 1) has three stages: (1) preprocessing (encoding) at the central controller, (2) parallel computing at $n > k$ parallel workers, and (3) post-processing (decoding) at the central controller. As we show later in the analysis of computing error, the entries $\text{trace}(\mathbf{C}(l_i))$ in the diagonal matrix $\mathbf{\Lambda}$ are the expected MSE at each worker prior to decoding. The decoding matrix $\mathbf{L}_{k \times n}$ in the decoding step (7) is chosen to be $(\mathbf{G}\mathbf{\Lambda}^{-1}\mathbf{G}^\top)^{-1}\mathbf{G}\mathbf{\Lambda}^{-1}$ to reduce the mean-squared error of the estimates of linear inverse solutions by assigning different weights to different workers based on the estimated accuracy of their computation (which is what $\mathbf{\Lambda}$ provides). This particular choice of $\mathbf{\Lambda}$ is inspired from the weighted least-square solution.

---

**Algorithm 1** Coded Distributed Linear Inverse

**Input:** Input vectors $[\mathbf{r}_1, \mathbf{r}_2, \ldots, \mathbf{r}_k]$, generator matrix $\mathbf{G}_{k \times n}$, the linear system matrices $\mathbf{B}$ and $\mathbf{K}$ defined in (1).
**Initialize (Encoding):** Encode the input vectors and the initial estimates by multiplying $\mathbf{G}$:

$$[\mathbf{s}_1, \mathbf{s}_2, \ldots, \mathbf{s}_n] = [\mathbf{r}_1, \mathbf{r}_2, \ldots, \mathbf{r}_k] \cdot \mathbf{G}. \tag{5}$$

$$[\mathbf{y}_1^{(0)}, \mathbf{y}_2^{(0)}, \ldots, \mathbf{y}_n^{(0)}] = [\mathbf{x}_1^{(0)}, \mathbf{x}_2^{(0)}, \ldots, \mathbf{x}_k^{(0)}] \cdot \mathbf{G}. \tag{6}$$

**Parallel Computing:**
**for** $i = 1$ **to** $n$ (in parallel) **do**
  Send $\mathbf{s}_i$ and $\mathbf{y}_i^{(0)}$ to the $i$-th worker. Execute the iterative method (1) with initial estimate $\mathbf{y}_i^{(0)}$ and input $\mathbf{s}_i$ at each worker.
**end for**
After a deadline time $T_\text{dl}$, collect all linear inverse results $\mathbf{y}_i^{(l_i)}$ from these $n$ workers. The superscript $l_i$ in $\mathbf{y}_i^{(l_i)}$ represents that the $i$-th worker finished $l_i$ iterations. Denote by $\mathbf{Y}^{(T_\text{dl})}$ the collection of all results $\mathbf{Y}_{N \times n}^{(T_\text{dl})} = [\mathbf{y}_1^{(l_1)}, \mathbf{y}_2^{(l_2)}, \ldots, \mathbf{y}_n^{(l_n)}]$.
**Post Processing (decoding at the central controller):**
Compute an estimate of the linear inverse solutions using the following matrix multiplication:

$$\hat{\mathbf{X}}^\top = \mathbf{L} \cdot (\mathbf{Y}^{(T_\text{dl})})^\top := (\mathbf{G}\mathbf{\Lambda}^{-1}\mathbf{G}^\top)^{-1}\mathbf{G}\mathbf{\Lambda}^{-1}(\mathbf{Y}^{(T_\text{dl})})^\top, \tag{7}$$

where the estimate $\hat{\mathbf{X}}_{N \times k} = [\hat{\mathbf{x}}_1, \hat{\mathbf{x}}_2, \ldots, \hat{\mathbf{x}}_k]$, the matrix $\mathbf{\Lambda}$ is

$$\mathbf{\Lambda} = \text{diag}\left[\text{trace}(\mathbf{C}(l_1)), \ldots, \text{trace}(\mathbf{C}(l_n))\right], \tag{8}$$

where the matrices $\mathbf{C}(l_i), i = 1, \ldots, n$ are defined as

$$\mathbf{C}(l_i) = \mathbf{B}^{l_i}\mathbf{C}_E(\mathbf{B}^\top)^{l_i}. \tag{9}$$

In computation of $\mathbf{\Lambda}$, if $\text{trace}(\mathbf{C}(l_i))$ are not available, one can use precomputed estimates of this trace as discussed in Supplementary Section 8.9 with negligible computational complexity and theoretically guaranteed accuracy.

---

## 3.1 Bounds on Performance of the Coded Linear Inverse Algorithm

Define $\mathbf{l} = [l_1, l_2, \ldots l_n]$ as the vector of the number of iterations at all workers. $\mathbb{E}[\cdot | \mathbf{l}]$ denotes the conditional expectation taken with respect to the randomness of the optimal solution $\mathbf{x}_i^*$ (see Assumption 1) conditioned on fixed iteration number $l_i$ at each worker, i.e., $\mathbb{E}[X | \mathbf{l}] = \mathbb{E}[X | l_1, l_2, \ldots l_n]$. Define $\mathbf{X}_{N \times k}^* = [\mathbf{x}_1^*, \mathbf{x}_2^*, \ldots \mathbf{x}_k^*]$ as the matrix composed of all the true solutions.

*Theorem* 3.1. Define $\mathbf{E} = \hat{\mathbf{X}} - \mathbf{X}^*$, i.e., the error of the decoding result (7). Assuming that the solutions for each linear inverse problem are chosen i.i.d. (across all problems) according to a distribution with covariance $\mathbf{C}_E$. Then, the error covariance of $\mathbf{E}$ satisfies

$$\mathbb{E}[\|\mathbf{E}\|^2 | \mathbf{l}] \leq \sigma_{\max}(\mathbf{G}^\top\mathbf{G})\text{trace}\left[(\mathbf{G}\mathbf{\Lambda}^{-1}\mathbf{G}^\top)^{-1}\right], \tag{10}$$

where the norm $\|\cdot\|$ is the Frobenius norm, $\sigma_{\max}(\mathbf{G}^\top\mathbf{G})$ is the maximum eigenvalue of $\mathbf{G}^\top\mathbf{G}$ and the matrix $\mathbf{\Lambda}$ is defined in (8). Further, when $\mathbf{G}$ has orthonormal rows,

$$\mathbb{E}[\|\mathbf{E}\|^2 | \mathbf{l}] \leq \text{trace}\left[(\mathbf{G}\mathbf{\Lambda}^{-1}\mathbf{G}^\top)^{-1}\right], \tag{11}$$

*Proof overview.* See supplementary Section 8.3 for the complete proof. Here we provide the main intuition by analyzing a "scalar version" of the linear inverse problem, in which case the matrix $\mathbf{B}$ is equal to a scalar $a$. For $\mathbf{B} = a$, the inputs and the initial estimates in (5) and (6) are vectors instead of matrices. As we show in Supplementary Section 8.3, if we encode both the inputs and the initial estimates using (5) and (6), we also "encode" the error

$$[\epsilon_1^{(0)}, \epsilon_2^{(0)}, \ldots, \epsilon_n^{(0)}] = [e_1^{(0)}, e_2^{(0)}, \ldots, e_k^{(0)}] \cdot \mathbf{G} =: \mathbf{E}_0 \mathbf{G}, \tag{12}$$

where $\epsilon_i^{(0)} = y_i^{(0)} - y_i^*$ is the initial error at the $i$-th worker, $e_i^{(0)} = x_i^{(0)} - x_i^*$ is the initial error of the $i$-th linear inverse problem, and $\mathbf{E}_0 := [e_1^{(0)}, e_2^{(0)}, \ldots e_k^{(0)}]$. Suppose $\text{var}[e_i^{(0)}] = c_e$, which is a scalar version of $\mathbf{C}_E$ after Assumption 1. From (4), the error satisfies:

$$\epsilon_i^{(l_i)} = a^{l_i} \epsilon_i^{(0)}, i = 1, 2, \ldots n. \tag{13}$$

Denote by $\mathbf{D} = \text{diag}\{a^{l_1}, a^{l_2}, \ldots a^{l_n}\}$. Therefore, from (12) and (13), the error before the decoding step (7) can be written as

$$[\epsilon_1^{(l_1)}, \epsilon_2^{(l_2)}, \ldots \epsilon_n^{(l_n)}] = [\epsilon_1^{(0)}, \epsilon_2^{(0)}, \ldots \epsilon_n^{(0)}] \cdot \mathbf{D} = \mathbf{E}_0 \mathbf{G} \mathbf{D}. \tag{14}$$

We can show (see Supplementary Section 8.3 for details) that after the decoding step (7), the error vector is also multiplied by the decoding matrix $\mathbf{L} = (\mathbf{G}\mathbf{\Lambda}^{-1}\mathbf{G}^\top)^{-1}\mathbf{G}\mathbf{\Lambda}^{-1}$:

$$\mathbf{E}^\top = \mathbf{L} \left[\epsilon_1^{(l_1)}, \epsilon_2^{(l_2)}, \ldots \epsilon_n^{(l_n)}\right]^\top = \mathbf{L}\mathbf{D}^\top\mathbf{G}^\top\mathbf{E}_0^\top. \tag{15}$$

Thus,

$$
\begin{aligned}
\mathbb{E}[\|\mathbf{E}\|^2 \,|\mathbf{l}] &= \mathbb{E}[\text{trace}[\mathbf{E}^\top\mathbf{E}]|\mathbf{l}] = \text{trace}[\mathbf{L}\mathbf{D}^\top\mathbf{G}^\top\mathbb{E}[\mathbf{E}_0^\top\mathbf{E}_0|\mathbf{l}]\mathbf{G}\mathbf{D}\mathbf{L}^\top] \\
&\overset{(a)}{=} \text{trace}[\mathbf{L}\mathbf{D}^\top\mathbf{G}^\top c_e \mathbf{I}_k \mathbf{G}\mathbf{D}\mathbf{L}^\top] = c_e \text{trace}[\mathbf{L}\mathbf{D}^\top\mathbf{G}^\top\mathbf{G}\mathbf{D}\mathbf{L}^\top] \\
&\overset{(b)}{\leq} c_e \sigma_{\max}(\mathbf{G}^\top\mathbf{G})\text{trace}[\mathbf{L}\mathbf{D}^\top\mathbf{D}\mathbf{L}^\top] = \sigma_{\max}(\mathbf{G}^\top\mathbf{G})\text{trace}[\mathbf{L}(c_e\mathbf{D}^\top\mathbf{D})\mathbf{L}^\top] \\
&\overset{(c)}{=} \sigma_{\max}(\mathbf{G}^\top\mathbf{G})\text{trace}[\mathbf{L}\mathbf{\Lambda}\mathbf{L}^\top] \overset{(d)}{=} \sigma_{\max}(\mathbf{G}^\top\mathbf{G})\text{trace}[(\mathbf{G}\mathbf{\Lambda}^{-1}\mathbf{G}^\top)^{-1}],
\end{aligned}
\tag{16}
$$

where (a) holds because $\mathbf{E}_0 := [e_1^{(0)}, e_2^{(0)}, \ldots e_k^{(0)}]$ and $\text{var}[e_i^{(0)}] = c_e$, (b) holds because $\mathbf{G}^\top\mathbf{G} \preceq \sigma_{\max}(\mathbf{G}^\top\mathbf{G})\mathbf{I}_n$, (c) holds because $c_e\mathbf{D}^\top\mathbf{D} = \mathbf{\Lambda}$, which is from the fact that for a scalar linear system matrix $\mathbf{B} = a$, the entries in the $\mathbf{\Lambda}$ matrix in (8) satisfy

$$\text{trace}(\mathbf{C}(l_i)) = a^{l_i}c_e(a^\top)^{l_i} = c_e a^{2l_i}, \tag{17}$$

which is the same as the entries in the diagonal matrix $c_e\mathbf{D}^\top\mathbf{D}$. Finally, (d) is obtained by directly plugging in $\mathbf{L} := (\mathbf{G}\mathbf{\Lambda}^{-1}\mathbf{G}^\top)^{-1}\mathbf{G}\mathbf{\Lambda}^{-1}$. Finally, inequality 11 holds because when $\mathbf{G}$ has orthonormal rows, $\sigma(\mathbf{G}^\top\mathbf{G}) = 1$.

Additionally, we note that in (10), the term $\text{trace}\left[(\mathbf{G}\mathbf{\Lambda}^{-1}\mathbf{G}^\top)^{-1}\right]$ resembles the MSE of ordinary weighted least-square solution, and the term $\sigma_{\max}(\mathbf{G}^\top\mathbf{G})$ represents the "inaccuracy" due to using the weighted least-square solution as the decoding result, because the inputs to different workers become correlated by multiplying the i.i.d. inputs with matrix $\mathbf{G}$ (see (5)). $\square$

## 4 Comparison with Uncoded Schemes and Replication-based Schemes

Here, we often assume (we will state explicitly in the theorem) that the number of iterations $l_i$ at different workers are i.i.d.. We use $\mathbb{E}_f[\cdot]$ to denote expectation on randomness of both the linear inverse solutions $\mathbf{x}_i^*$ and the number of iterations $l_i$ (this is different from the notation $\mathbb{E}[\cdot|\mathbf{l}]$).

*Assumption* 2. Within time $T_{\text{dl}}$, the number of iterations of linear inverse computations (see (1)) at each worker follows an i.i.d. distribution $l_i \sim f(l)$.

### 4.1 Comparison between the coded and uncoded linear inverse before a deadline

First, we compare the coded linear inverse scheme with an uncoded scheme, in which case we use the first $k$ workers to solve $k$ linear inverse problems in (2) without coding. The following theorem quantifies the overall mean-squared error of the uncoded scheme given $l_1, l_2, \ldots, l_k$. The proof is in Supplementary Section 8.6.

*Theorem* 4.1. In the uncoded scheme, the error $\mathbb{E}\left[\left\|\mathbf{E}_{\text{uncoded}}\right\|^2 |\mathbf{l}\right] = \mathbb{E}\left[\left\|[\mathbf{e}_1^{(l_1)} \ldots, \mathbf{e}_k^{(l_k)}]\right\|^2 \bigg| \mathbf{l}\right] = \sum_{i=1}^{k} \text{trace}\left(\mathbf{C}(l_i)\right)$. Further, when the i.i.d. Assumption 2 holds,

$$\mathbb{E}_f\left[\left\|\mathbf{E}_{\text{uncoded}}\right\|^2\right] = k\mathbb{E}_f[\text{trace}(\mathbf{C}(l_1))]. \tag{18}$$

Then, we compare the overall mean-squared error of coded and uncoded linear inverse algorithms. *Note that this comparison is not fair because the coded algorithm uses more workers than uncoded. However, we still include Theorem 4.2 because we need it for the fair comparison between coded and replication-based linear inverse.* The proof is in Supplementary section 8.4.

*Theorem* 4.2. (Coded linear inverse beats uncoded) Suppose the i.i.d. Assumptions 1 and 2 hold and suppose $\mathbf{G}$ is a $k \times n$ submatrix of an $n \times n$ Fourier transform matrix $\mathbf{F}$, i.e., $\mathbf{F}_{n \times n} = \begin{bmatrix} \mathbf{G}_{k \times n} \\ \mathbf{H}_{(n-k) \times n} \end{bmatrix}$. Then, expected error of the coded linear inverse is strictly less than that of uncoded:

$$\mathbb{E}_f\left[\left\|\mathbf{E}_{\text{uncoded}}\right\|^2\right] - \mathbb{E}_f\left[\left\|\mathbf{E}_{\text{coded}}\right\|^2\right] \geq \mathbb{E}_f[\text{trace}(\mathbf{J}_2\mathbf{J}_4^{-1}\mathbf{J}_2^\top)], \tag{19}$$

where $\mathbf{J}_2$ and $\mathbf{J}_4$ are the submatrices of $\mathbf{F}\boldsymbol{\Lambda}\mathbf{F}^\top := \begin{bmatrix} \mathbf{J}_1 & \mathbf{J}_2 \\ \mathbf{J}_2^\top & \mathbf{J}_4 \end{bmatrix}_{n \times n}$ and the matrix $\boldsymbol{\Lambda}$ is defined in (8). That is, $(\mathbf{J}_1)_{k \times k}$ is $\mathbf{G}\boldsymbol{\Lambda}\mathbf{G}^\top$, $(\mathbf{J}_2)_{k \times (n-k)}$ is $\mathbf{G}\boldsymbol{\Lambda}\mathbf{H}^\top$, and $(\mathbf{J}_4)_{(n-k) \times (n-k)}$ is $\mathbf{H}\boldsymbol{\Lambda}\mathbf{H}^\top$.

## 4.2 Comparison between the replication-based and coded linear inverse before a deadline

Consider an alternative way of doing linear inverse using $n > k$ workers. In this paper, we only consider the case when $n - k < k$, i.e., the number of extra workers is only slightly bigger than the number of problems (both in theory and in experiments). Since we have $n - k$ extra workers, a natural way is to pick any $(n - k)$ linear inverse problems and replicate them using these extra $(n - k)$ workers. After we obtain two computation results for the same equation, we use two natural "decoding" strategies for this replication-based linear inverse: (i) choose the worker with higher number of iterations; (ii) compute the weighted average using weights $\frac{w_1}{w_1+w_2}$ and $\frac{w_2}{w_1+w_2}$, where $w_1 = 1/\sqrt{\text{trace}(\mathbf{C}(l_1))}$ and $w_2 = 1/\sqrt{\text{trace}(\mathbf{C}(l_2))}$, and $l_1$ and $l_2$ are the number of iterations completed at the two workers (recall that $\text{trace}(\mathbf{C}(l_i))$ represents the residual MSE at the $i$-th worker).

*Theorem* 4.3. The replication-based schemes satisfy the following lower bound on the MSE:

$$\mathbb{E}_f\left[\left\|\mathbf{E}_{\text{rep}}\right\|^2\right] > \mathbb{E}_f\left[\left\|\mathbf{E}_{\text{uncoded}}\right\|^2\right] - (n - k)\mathbb{E}_f[\text{trace}(\mathbf{C}(l_1))]. \tag{20}$$

*Proof overview.* Here the goal is to obtain a lower bound on the MSE of replication-based linear inverse and compare it with an upper bound on the MSE of coded linear inverse.

Note that if an extra worker is used to replicate the computation at the $i$-th worker, i.e., the linear inverse problem with input $\mathbf{r}_i$ is solved on two workers, the expected error of the result of the $i$-th problem could best reduced from $\mathbb{E}_f[\text{trace}(\mathbf{C}(l_1))]$ (see Thm. 4.1) to zero[1]. Therefore, $(n-k)$ extra workers make the error decrease by at most (and strictly smaller than) $(n - k)\mathbb{E}_f[\text{trace}(\mathbf{C}(l_1))]$. □

Using this lower bound, we can provably show that coded linear inverse beats replication-based linear inverse when certain conditions are satisfied. One crucial condition is that the distribution of the random variable $\text{trace}(\mathbf{C}(l))$ (i.e., the expected MSE at each worker) satisfies a "variance heavy-tail" property defined as follows.

*Definition* 1. The random variable $\text{trace}(\mathbf{C}(l))$ is said to have a "$\rho$-variance heavy-tail" property if

$$\text{var}_f[\text{trace}(\mathbf{C}(l))] > \rho\mathbb{E}_f^2[\text{trace}(\mathbf{C}(l))], \tag{21}$$

for some constant $\rho > 1$. Notice that the term $\text{trace}(\mathbf{C}(l))$ is essentially the remaining MSE after $l$ iterations at a single machine. Therefore, this property simply means the remaining error at a single machine has large variance. For the coded linear inverse, we will use a "Fourier code", the generator matrix $\mathbf{G}$ of which is a submatrix of a Fourier matrix. This particular choice of code is only for ease of analysis in comparing coded linear inverse and replication-based linear inverse. In practice, the code that minimizes mean-squared error should be chosen.

*Theorem* 4.4. (Coded linear inverse beats replication) Suppose the i.i.d. Assumptions 1 and 2 hold and $\mathbf{G}$ is a $k \times n$ submatrix of $k$ rows of an $n \times n$ Fourier matrix $\mathbf{F}$. Further, suppose $(n - k) = o(\sqrt{n})$. Then, the expected error of the coded linear inverse satisfies

$$\lim_{n \to \infty} \frac{1}{n-k} \left[ \mathbb{E}_f \left[ \|\mathbf{E}_{\text{uncoded}}\|^2 \right] - \mathbb{E}_f \left[ \|\mathbf{E}_{\text{coded}}\|^2 \right] \right] \geq \frac{\text{var}_f[\text{trace}(\mathbf{C}(l_1))]}{\mathbb{E}_f[\text{trace}(\mathbf{C}(l_1))]}. \tag{22}$$

Moreover, if the random variable $\text{trace}(\mathbf{C}(l))$ satisfies the $\rho$-variance heavy-tail property for $\rho > 1$, coded linear inverse outperforms replication-based linear inverse in the following sense,

$$\lim_{n \to \infty} \frac{1}{(n-k)} \left[ \mathbb{E}_f \left[ \|\mathbf{E}_{\text{uncoded}}\|^2 \right] - \mathbb{E}_f \left[ \|\mathbf{E}_{\text{rep}}\|^2 \right] \right] < \frac{1}{\rho} \lim_{n \to \infty} \frac{1}{(n-k)} \left[ \mathbb{E}_f \left[ \|\mathbf{E}_{\text{uncoded}}\|^2 \right] - \mathbb{E}_f \left[ \|\mathbf{E}_{\text{coded}}\|^2 \right] \right]. \tag{23}$$

*Proof overview.* See Supplementary Section 8.7 for a complete and rigorous proof. **Here we only provide the main intuition behind the proof.** From Theorem 4.2, we have $\mathbb{E}_f \left[ \|\mathbf{E}_{\text{uncoded}}\|^2 \right] - \mathbb{E}_f \left[ \|\mathbf{E}_{\text{coded}}\|^2 \right] \geq \mathbb{E}_f[\text{trace}(\mathbf{J}_2 \mathbf{J}_4^{-1} \mathbf{J}_2^{\top})]$. Therefore, to prove (22), the main technical difficulty is to simplify the term $\text{trace}(\mathbf{J}_2 \mathbf{J}_4^{-1} \mathbf{J}_2^{\top})$. For a Fourier matrix $\mathbf{F}$, we are able to show that the matrix $\mathbf{F} \mathbf{\Lambda} \mathbf{F}^{\top} = \begin{bmatrix} \mathbf{J}_1 & \mathbf{J}_2 \\ \mathbf{J}_2^{\top} & \mathbf{J}_4 \end{bmatrix}$ (see Theorem 4.2) is a Toeplitz matrix, which provides a good structure for us to study its behavior. Then, we use the Gershgorin circle theorem [27] (with some algebraic manipulations) to show that the maximum eigenvalue of $\mathbf{J}_4$ satisfies $\sigma_{\max}(\mathbf{J}_4) \approx \mathbb{E}_f[\text{trace}(\mathbf{C}(l_1))]$, and separately using some algebraic manipulations, we show

$$\text{trace}(\mathbf{J}_2 \mathbf{J}_2^{\top}) \approx (n-k) \text{var}_f[\text{trace}(\mathbf{C}(l_1))], \tag{24}$$

for large matrix size $n$. Since $\text{trace}(\mathbf{J}_2 \mathbf{J}_4^{-1} \mathbf{J}_2^{\top}) \geq \text{trace}(\mathbf{J}_2 (\sigma_{\max}(\mathbf{J}_4))^{-1} \mathbf{J}_2^{\top}) = \frac{1}{\sigma_{\max}(\mathbf{J}_4)} \text{trace}(\mathbf{J}_2 \mathbf{J}_2^{\top})$,

$$\text{trace}(\mathbf{J}_2 \mathbf{J}_4^{-1} \mathbf{J}_2^{\top}) \geq \frac{(n-k) \text{var}_f[\text{trace}(\mathbf{C}(l_1))]}{\mathbb{E}_f[\text{trace}(\mathbf{C}(l_1))]}, \tag{25}$$

for large $n$. Then, (22) can be proved by plugging (25) into (19). After that, we can combine (22), (20) and the variance heavy-tail property to prove (23). $\qquad \square$

### 4.3 Asymptotic Comparison between Coded, Uncoded and Replication-based linear inverse as the Deadline $T_{\text{dl}} \to \infty$

*Assumption* 3. We assume the computation time of one power iteration is fixed at each worker for each linear inverse computation, i.e., there exist $n$ independent (not necessarily identically distributed) random variables $v_1, v_2, \ldots v_n$ such that $l_i = \lceil \frac{T_{\text{dl}}}{v_i} \rceil, i = 1, 2, \ldots n$.

The above assumption is validated in experiments in Supplementary Section 8.13.

The $k$-th order statistic of a sample is equal to its $k$-th smallest value. Suppose the order statistics of the sequence $v_1, v_2, \ldots v_n$ are $v_{i_1} < v_{i_2} < \ldots v_{i_n}$, where $\{i_1, i_2, \ldots i_n\}$ is a permutation of $\{1, 2, \ldots n\}$. Denote by $[k]$ the set $\{1, 2, \ldots k\}$ and $[n]$ the set $\{1, 2, \ldots n\}$.

*Theorem* 4.5. (Error exponent comparison when $T_{\text{dl}} \to \infty$) Suppose the i.i.d. Assumption 1 and Assumption 3 hold. Suppose $n - k < k$. Then, the error exponents of the coded and uncoded computation schemes satisfy

$$\lim_{T_{\text{dl}} \to \infty} -\frac{1}{T_{\text{dl}}} \log \mathbb{E}[\|\mathbf{E}_{\text{coded}}\|^2 \,|\mathbf{l}] \geq \frac{2}{v_{i_k}} \log \frac{1}{1-d}, \tag{26}$$

$$\lim_{T_{\text{dl}} \to \infty} -\frac{1}{T_{\text{dl}}} \log \mathbb{E}[\|\mathbf{E}_{\text{uncoded}}\|^2 \,|\mathbf{l}] = \lim_{T_{\text{dl}} \to \infty} -\frac{1}{T_{\text{dl}}} \log \mathbb{E}[\|\mathbf{E}_{\text{rep}}\|^2 \,|\mathbf{l}] = \frac{2}{\max_{i \in [k]} v_i} \log \frac{1}{1-d}. \tag{27}$$

The error exponents satisfy coded>replication=uncoded. Here the expectation $\mathbb{E}[\cdot|\mathbf{l}]$ is only taken with respect to the randomness of the linear inverse sequence $\mathbf{x}_i, i = 1, 2, \ldots k$.

*Proof overview.* See Supplementary Section 8.8 for a detailed proof. The main intuition behind this result is the following: when $T_{\text{dl}}$ approaches infinity, the error of uncoded computation is dominated

by the slowest worker among the first $k$ workers, which has per-iteration time $\max_{i \in [k]} v_i$. For the replication-based scheme, since the number of extra workers $n-k < k$, there is a non-zero probability (which does not change with $T_{\text{dl}}$) that the $n-k$ extra workers do not replicate the computation in the slowest one among the first worker. Therefore, replication when $n-k < k$ does not improve the error exponent, because the error is dominated by this slowest worker. For coded computation, we show in Supplementary Section 8.8 that the slowest $n-k$ workers among the overall $n$ workers do not affect the error exponent, which means that the error is dominated by the $k$-th fastest worker, which has per-iteration time $v_{i_k}$. Since the $k$-th fastest worker among all $n$ workers can not be slower than the slowest one among the first (unordered) $k$ workers, the error exponent of coded linear inverse is larger than that of the uncoded and the replication-based linear inverse. □

## 5 Analyzing the Computational Complexity

### 5.1 Encoding and decoding complexity

We first show that the encoding and decoding complexity of Algorithm 1 are in scaling-sense smaller than that of the computation at each worker. This ensures that straggling comes from the parallel workers, not the encoder or decoder. The proof of Theorem 5.1 is in Supplementary Section 8.10. In our experiment on the Google Plus graph (See Section 6) for computing PageRank, the computation time at each worker is 30 seconds and the encoding and decoding time at the central controller is about 1 second.

*Theorem* 5.1. The computational complexity for the encoding and decoding is $\Theta(nkN)$, where $N$ is the number of rows in the matrix $\mathbf{B}$ and $k, n$ depend on the number of available workers assuming that each worker performs a single linear inverse computation. For a general dense matrix $\mathbf{B}$, the computational complexity of computing linear inverse at each worker is $\Theta(N^2 l)$, where $l$ is the number of iterations in the specified iterative algorithm. The complexity of encoding and decoding is smaller than that of the computation at each user for large $\mathbf{B}$ matrices (large $N$).

### 5.2 Analysis on the cost of communication versus computation

In this work, we focus on optimizing the computation cost. However, what if the computation cost is small compared to the overall cost, including the communication cost? If this is true, optimizing the computation cost is not very useful. In Theorem 5.2 (proof appears in Supplementary Section 8.11), we show that the computation cost is larger than the communication cost in the scaling-sense.

*Theorem* 5.2. The ratio between the number of operations (computation) and the number of bits transmitted (communication) at the $i$-th worker is $\text{COST}_{\text{computation}}/\text{COST}_{\text{communication}} = \Theta(l_i \bar{d})$ operations per integer, where $l_i$ is the number of iterations at the $i$-th worker, and $\bar{d}$ is the average number of non-zeros in each row of the $\mathbf{B}$ matrix.

## 6 Experiments on Real Systems

We test the performance of the coded linear inverse algorithm for the PageRank problem on the Twitter graph and the Google Plus graph from the SNAP datasets [28]. The Twitter graph has 81,306 nodes and 1,768,149 edges, and Google Plus graph has 107,614 nodes and 13,673,453 edges. We use the HT-condor framework in a cluster to conduct the experiments. The task is to solve $k = 100$ personalized PageRank problems in parallel using $n = 120$ workers. The uncoded algorithm picks the first $k$ workers and uses one worker for each PageRank problem. The two replication-based schemes replicate the computation of the first $n-k$ PageRank problems in the extra $n-k$ workers (see Section 4.2). The coded PageRank uses $n$ workers to solve these $k = 100$ equations using Algorithm 1. We use a $(120, 100)$ code where the generator matrix is the submatrix composed of the first 100 rows in a $120 \times 120$ DFT matrix. The computation results are shown in the left two figures in Fig. 2. Note that the two graphs are of different sizes so the computation in the two experiments take different time. From Fig. 2, we can see that the mean-squared error of uncoded and replication-based schemes is larger than that of coded computation by a factor of $10^4$ for large deadlines.

We also compare Algorithm 1 with the coded computing algorithm proposed in [6]. As we discussed in the Figure 1, the original coded technique in [6] ignores partial results and is suboptimal even in the toy example of three workers. However, it has a natural extension to iterative methods, which will be

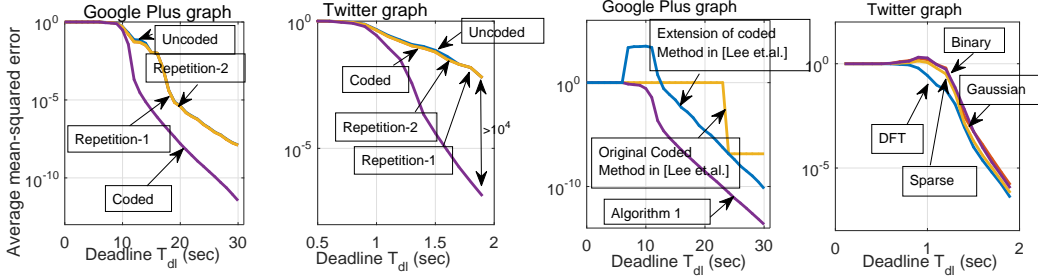

Figure 2: From left to right: (1,2) Experimentally computed overall MSE of uncoded, replication-based and coded personalized PageRank on the Twitter and Google Plus graph on a cluster with 120 workers. The ratio of MSE for repetition-based schemes and coded linear inverse increase as $T_{dl}$ increases. (3) Comparison between an extended version of the algorithm in [6] and Algorithm 1 on the Google Plus graph. The figure shows that naively extending the general coded method using matrix inverse introduces error amplification. (4) Comparison of different codes. In this experiment the DFT-code out-performs the other candidates in MSE.

discussed in details later. The third figure in Fig. 2 shows the comparison between the performance of Algorithm 1 and this extension of the algorithm from [6]. This extension uses the (unfinished) partial results from the $k$ fastest workers to retrieve the required PageRank solutions. More concretely, suppose $\mathcal{S} \subset [n]$ is the index set of the $k$ fastest workers. Then, this extension retrieves the solutions to the original $k$ PageRank problems by solving the equation $\mathbf{Y}_{\mathcal{S}} = [\mathbf{x}_1^*, \mathbf{x}_2^*, \ldots, \mathbf{x}_k^*] \cdot \mathbf{G}_{\mathcal{S}}$, where $\mathbf{Y}_{\mathcal{S}}$ is composed of the (partial) computation results obtained from the fastest $k$ workers and $\mathbf{G}_{\mathcal{S}}$ is the $k \times k$ submatrix composed of the columns in the generator matrix $\mathbf{G}$ with indexes in $\mathcal{S}$. However, since there is some remaining error at each worker (i.e., the computation results $\mathbf{Y}_{\mathcal{S}}$ have not converged yet), when conducting the matrix-inverse-based decoding from [6], the error is magnified due to the large condition number of $\mathbf{G}_{\mathcal{S}}$. This is why the algorithm in [6] should not be naively extended in the coded linear inverse problem.

One question remains: what is the best code design for the coded linear inverse algorithm? Although we do not have a concrete answer to this question, we have tested different codes (with different generator matrices $\mathbf{G}$) in the Twitter graph experiment, all using Algorithm 1. The results are shown in the fourth figure in Fig. 2. The generator matrix used for the "binary" curve has i.i.d. binary entries in $\{-1, 1\}$. The generator matrix used for the "sparse" curve has random binary sparse entries. The generator matrix for the "Gaussian" curve has i.i.d. standard Gaussian entries. In this experiment, the DFT-code performs the best. However, finding the best code in general is a meaningful future work.

## 7 Conclusions

By studying coding for iterative algorithms designed for distributed inverse problems, we aim to introduce new applications and analytical tools to the problem of coded computing with stragglers. Since these iterative algorithms designed for inverse problems commonly have decreasing error with time, the partial computation results at stragglers can provide useful information for the final outputs. Note that this is unlike recent works on coding for multi-stage computing problems [29, 30], where the computation error can accumulate with time and coding has to be applied repeatedly to suppress this error accumulation. An important connection worth discussing is the *diversity gain* in this coded computing problem. The distributed computing setting in this work resembles random fading channels, which means coding can be used to exploit straggling diversity just as coding is used in communication channels to turn diverse channel fading into an advantage. What makes coding even more suitable in our setting is that the amount of diversity gain achieved here through *replication* is actually smaller than that can be achieved by replication in fading channels. This is because for two computers that solve the same equation $\mathbf{M}\mathbf{x}_i = \mathbf{r}_i$, the remaining error at the slow worker is a deterministic multiple of the remaining error at the fast worker (see equation (3)). Therefore, taking a weighted average of the two computation results through replication does not reduce error as in independent fading channels. How diversity gain can be achieved here optimally is worth deep investigation. Our next goals are two-fold: (1) extend the current method to solving a single large-scale inverse problem, such as graph mining with graphs that exceed the memory of a single machine; (2) carry out experiments on faster distributed systems such as Amazon EC2.

## Footnotes

[1]Although this is clearly a loose bound, it makes for convenient comparison with coded linear inverse.

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
