[Supplementary Material]

# 8 Supplementary Materials

## 8.1 Graph Signal Sampling and Recovery as a Motivating Example for solving Non-square Linear Inverse Problems using Iterative Methods

The emerging field of signal processing on graphs [25, 26] is based on an interesting idea of treating "values" associated with the nodes of a graph as "a signal supported on a graph" and apply techniques from signal processing to solve problems such as prediction and detection. For example, the number of cars at different road intersections on the Manhattan road network can be viewed as a graph signal on the road graph $\mathcal{G} = (\mathcal{V}, \mathcal{E})$ where $\mathcal{V}$ is the set of road intersections and $\mathcal{E}$ is the set of road segments. For a directed or undirected graph $\mathcal{G} = (\mathcal{V}, \mathcal{E})$, the graph signal has the same dimension as the number of nodes $|\mathcal{V}|$ in the graph, i.e., there is only one value associated with each node.

One important problem of graph signal processing is that of recovering the values on the remaining nodes of a graph given the values sampled on a particular node subset $\mathcal{S} \subset \mathcal{V}$ under the assumption of "bandlimited" graph signal [31–33]. One application of graph signal reconstruction can be that of reconstructing the entire traffic flow using the observations from a few cameras at some road intersections [34]. The graph signal reconstruction problem can be formulated as a least-square solution of the following linear system (see equation (5) in [32])

$$\begin{bmatrix} \mathbf{f}(\mathcal{S}) \\ \mathbf{f}(\mathcal{S}^c) \end{bmatrix} = \begin{bmatrix} \mathbf{u}_1(\mathcal{S}) & \mathbf{u}_2(\mathcal{S}) & \ldots & \mathbf{u}_w(\mathcal{S}) \\ \mathbf{u}_1(\mathcal{S}^c) & \mathbf{u}_2(\mathcal{S}^c) & \ldots & \mathbf{u}_w(\mathcal{S}^c) \end{bmatrix} \begin{bmatrix} \alpha_1 \\ \alpha_2 \\ \vdots \\ \alpha_w. \end{bmatrix}, \tag{28}$$

where $\mathbf{f}(\mathcal{S})$ is the given part of the graph signal $\mathbf{f}$ on the set $\mathcal{S}$, $\mathbf{f}(\mathcal{S}^c)$ is the unknown part of the graph signal $\mathbf{f}$ to be reconstructed, $[\alpha_1, \alpha_2, \ldots \alpha_w]$ are unknown coefficients of the graph signal $\mathbf{f} = \begin{bmatrix} \mathbf{f}(\mathcal{S}) \\ \mathbf{f}(\mathcal{S}^c) \end{bmatrix}$ represented in the subspace spanned by $\mathbf{u}_1, \mathbf{u}_2, \ldots \mathbf{u}_w$ which are the first $w$ eigenvectors of the graph Laplacian matrix. It is shown in [32] that this least-square reconstruction problem can be solved using an iterative linear projection method (see equation (11) in [32])

$$\mathbf{f}_{k+1} = \mathbf{U}\mathbf{U}^\top \left( \mathbf{f}_k + \mathbf{J}^\top \mathbf{J}\left( \begin{bmatrix} \mathbf{f}(\mathcal{S}) \\ \mathbf{0} \end{bmatrix} - \mathbf{f}_k \right) \right), \tag{29}$$

where $\mathbf{U} = [\mathbf{u}_1, \mathbf{u}_2, \ldots \mathbf{u}_w]$. This iteration can be shown to converge to the least square solution given by

$$\mathbf{f}(\mathcal{S}^c) = (\mathbf{U})_{\mathcal{S}^c}((\mathbf{U})_{\mathcal{S}^c}^\top (\mathbf{U})_{\mathcal{S}^c})^{-1}(\mathbf{U})_{\mathcal{S}^c}^t \mathbf{f}(\mathcal{S}). \tag{30}$$

Note that we may want to reconstruct multiple instances of graph signal, such as road traffic at different time, which brings in the formulation of distributed computing as discussed in Section 2.1.

## 8.2 Bounds on the Mean-squared Error beyond the i.i.d. Case

Until now, we based our analysis on the i.i.d. assumption 1. For the PageRank problem discussed in Section 2.2, this assumption means that the personalized PageRank queries (the different preference vector $\mathbf{r}_i$'s) are independent across different users. Although the case when the PageRank queries are arbitrarily correlated is hard to analyze, we may still provide concrete analysis for some specific cases. For example, a reasonable case when the PageRank queries are correlated with each other is when these queries are all affected by some "common fashion topic" that the users wish to search for. In mathematics, we can model this phenomenon by assuming that the solutions to the $i$-th linear inverse problem satisfies

$$\mathbf{x}_i^* = \bar{\mathbf{x}} + \mathbf{z}_i, \tag{31}$$

for some random vector $\bar{\mathbf{x}}$ and an i.i.d. vector $\mathbf{z}_i$ across different queries (different $i$). The common part $\bar{\mathbf{x}}$ is random because the common fashion topic itself can be random. This model can be generalized to the following "stationary" model.

*Assumption* 4. Assume the solutions $\mathbf{x}_i^*$'s of the linear inverse problems have the same mean $\boldsymbol{\mu}_E$ and stationary covariances, i.e.,

$$\mathbb{E}[\mathbf{x}_i^*(\mathbf{x}_i^*)^\top] = \mathbf{C}_E + \mathbf{C}_{\text{Cor}}, \forall 1 \leq i \leq k, \tag{32}$$

$$\mathbb{E}[\mathbf{x}_i^*(\mathbf{x}_j^*)^\top] = \mathbf{C}_{\text{Cor}}, \forall 1 \leq i, j \leq k. \tag{33}$$

Figure 3: Experimentally computed overall mean squared error of uncoded, replication-based and coded personalized PageRank on the Google Plus graph on a cluster with 120 workers. The queries are generated using the model from the stationary model in Assumption 4.

Under this assumption, we have to change the coded linear inverse algorithm slightly. The details are shown in Algorithm 2.

---

**Algorithm 2** Coded Distributed Linear Inverse (Stationary Inputs)

---

Call Algorithm 1 but replace the $\mathbf{\Lambda}$ matrix with

$$\tilde{\mathbf{\Lambda}} = \sigma_{\max}(\mathbf{G}^\top \mathbf{G})\mathbf{\Lambda} + \text{diag}\{\mathbf{G}^\top \mathbf{1}_k\} \cdot \mathbf{\Psi} \cdot \text{diag}\{\mathbf{G}^\top \mathbf{1}_k\}^\top, \tag{34}$$

where $\sigma_{\max}(\mathbf{G}^\top \mathbf{G})$ is the maximum eigenvalue of $\mathbf{G}^\top \mathbf{G}$, and $\mathbf{\Psi}_{n \times n} = [\Psi_{i,j}]$ satisfies

$$\Psi_{i,j} = \text{trace}[\mathbf{B}^{l_i} \mathbf{C}_{\text{cor}}(\mathbf{B}^\top)^{l_j}]. \tag{35}$$

---

For the stationary version, we can have the counterpart of Theorem 3.1 as follows. Trivial generalizations include arbitrary linear scaling $\mathbf{x}_i^* = \alpha_i \bar{\mathbf{x}} + \beta_i \mathbf{z}_i$ for scaling constants $\alpha_i$ and $\beta_i$.

*Theorem* 8.1. Define $\mathbf{E} = \hat{\mathbf{X}} - \mathbf{X}^*$, i.e., the error of the decoding result (7) by replacing $\mathbf{\Lambda}$ defined in (8) with $\tilde{\mathbf{\Lambda}}$ in (34). Assuming that the solutions for all linear inverse problems satisfy Assumption 4. Then, the error covariance of $\mathbf{E}$ satisfies

$$\mathbb{E}[\|\mathbf{E}\|^2 \,|\mathbf{l}] \leq \text{trace}\left[(\mathbf{G}\tilde{\mathbf{\Lambda}}^{-1}\mathbf{G}^\top)^{-1}\right]. \tag{36}$$

where the norm $\|\cdot\|$ is the Frobenius norm.

*Proof.* See Supplementary Section 8.5. $\square$

In Section 4, we have compared coded, uncoded and replication-based linear inverse schemes under the i.i.d. assumption. Now, we test Algorithm 2 for correlated PageRank queries that are distributed with the stationary covariance matrix in the form of (32) and (33). Note that the only change to be made in this case is on the $\mathbf{\Lambda}$ matrix (see equation (34)). The other settings are exactly the same as the experiments that are shown in Figure 2. The results on the Google Plus social graph are shown in Figure 3. In this case, we also have to compute the $\mathbf{\Lambda}$ matrix. This issue is discussed in Section 8.9.

## 8.3 Proof of Theorem 3.1

### 8.3.1 Notation and Preliminary Properties

Now we prove Theorem 3.1. We first introduce some notation and preliminary properties that we will use in this proof. Denote by $\text{vec}(\mathbf{A})$ the vector that is composed of the concatenation of all columns in a matrix $\mathbf{A}$. For example, the vectorization of $A = \begin{bmatrix} 1 & 2 & 3 \\ 4 & 5 & 6 \end{bmatrix}$ is the column vector

$\text{vec}(\mathbf{A}) = [1, 4, 2, 5, 3, 6]^\top$. We will also use the Kronecker product defined as

$$\mathbf{A}_{m \times n} \otimes \mathbf{B} = \begin{bmatrix} a_{11}\mathbf{B} & a_{12}\mathbf{B} & \dots & a_{1n}\mathbf{B} \\ \vdots & \vdots & \ddots & \vdots \\ a_{m1}\mathbf{B} & a_{m2}\mathbf{B} & \dots & a_{mn}\mathbf{B}. \end{bmatrix} \tag{37}$$

We now state some properties of the vectorization and Kronecker product.
*Lemma* 8.1. Property 1: if $\mathbf{A} = \mathbf{BC}$, then

$$\text{vec}(\mathbf{A}) = (\mathbf{C} \otimes \mathbf{I}_N)\text{vec}(\mathbf{B}). \tag{38}$$

Property 2: vectorization does not change the Frobenius norm, i.e.,

$$\|\mathbf{A}\| = \|\text{vec}(\mathbf{A})\|. \tag{39}$$

Property 3: The following mixed-product property holds

$$(\mathbf{A} \otimes \mathbf{B})(\mathbf{C} \otimes \mathbf{D}) = (\mathbf{A} \cdot \mathbf{C}) \otimes (\mathbf{B} \cdot \mathbf{D}), \tag{40}$$

if one can form the matrices $\mathbf{AC}$ and $\mathbf{BD}$.
Property 4: If $\mathbf{A}$ and $\mathbf{B}$ are both positive semi-definite, $\mathbf{A} \otimes \mathbf{B}$ is also positive semi-definite.
Property 5: Suppose $\mathbf{C}$ is positive semi-definite and $\mathbf{A} \preceq \mathbf{B}$. Then,

$$\mathbf{A} \otimes \mathbf{C} \preceq \mathbf{B} \otimes \mathbf{C}. \tag{41}$$

Property 6: (commutative property) Suppose $\mathbf{A}_{m \times n}$ and $\mathbf{B}_{p \times q}$ are two matrices. Then,

$$(\mathbf{A}_{m \times n} \otimes \mathbf{I}_p) \cdot (\mathbf{I}_n \otimes \mathbf{B}_{p \times q}) = (\mathbf{I}_m \otimes \mathbf{B}_{p \times q}) \cdot (\mathbf{A}_{m \times n} \otimes \mathbf{I}_q). \tag{42}$$

Property 7: Suppose $\mathbf{A}$ is an $nN \times nN$ matrix that can be written as

$$\mathbf{A}_{nN \times nN} = \begin{bmatrix} \mathbf{A}_{11} & \mathbf{A}_{12} & \dots & \mathbf{A}_{1n} \\ \mathbf{A}_{21} & \mathbf{A}_{22} & \dots & \mathbf{A}_{2n} \\ \vdots & \vdots & \ddots & \vdots \\ \mathbf{A}_{n1} & \mathbf{A}_{n2} & \dots & \mathbf{A}_{nn} \end{bmatrix}, \tag{43}$$

where each $\mathbf{A}_{ij}$ is a square matrix of size $N \times N$. Then, for an arbitrary matrix $\mathbf{L}$ of size $k \times n$,

$$\begin{aligned} &\text{trace}\left[(\mathbf{L} \otimes \mathbf{I}_N) \cdot \mathbf{A} \cdot (\mathbf{L} \otimes \mathbf{I}_N)^\top\right] \\ =&\text{trace}\left[\mathbf{L} \cdot \begin{bmatrix} \text{trace}[\mathbf{A}_{11}] & \dots & \text{trace}[\mathbf{A}_{1n}] \\ \vdots & \ddots & \vdots \\ \text{trace}[\mathbf{A}_{n1}] & \dots & \text{trace}[\mathbf{A}_{nn}] \end{bmatrix} \cdot \mathbf{L}^\top\right]. \end{aligned} \tag{44}$$

*Proof.* See Supplementary Section 8.14. □

### 8.3.2 Computing the explicit form of the error matrix E

From (5), we have encoded the input $\mathbf{r}_i$ to the linear inverse problem in the following way:

$$[\mathbf{s}_1, \mathbf{s}_2, \dots, \mathbf{s}_n] = [\mathbf{r}_1, \mathbf{r}_2, \dots, \mathbf{r}_k] \cdot \mathbf{G}. \tag{45}$$

Since $\mathbf{x}_i^*$ is the solution to the linear inverse problem, we have

$$\mathbf{x}_i^* = \mathbf{C}_{\text{inv}}\mathbf{r}_i, \tag{46}$$

where $\mathbf{C}_{\text{inv}}$ is either the direct inverse $\mathbf{M}^{-1}$ for square linear inverse problems or the least-square matrix $(\mathbf{M}^\top\mathbf{M} + \lambda I)^{-1}\mathbf{M}^\top$ for non-square inverse problems. Define $\mathbf{y}_i^*$ as the solution of the inverse problem with the encoded input $\mathbf{s}_i$. Then, we also have

$$\mathbf{y}_i^* = \mathbf{C}_{\text{inv}}\mathbf{s}_i. \tag{47}$$

Left-multiplying $\mathbf{C}_{\text{inv}}$ on both LHS and RHS of (45) and plugging in (46) and (47), we have

$$[\mathbf{y}_1^*, \mathbf{y}_2^*, \dots, \mathbf{y}_n^*] = [\mathbf{x}_1^*, \mathbf{x}_2^*, \dots, \mathbf{x}_k^*] \cdot \mathbf{G} = \mathbf{X}^* \cdot \mathbf{G}. \tag{48}$$

Define $\epsilon_i^{(l)} = \mathbf{y}_i^{(l_i)} - \mathbf{y}_i^*$, which is the remaining error at the $i$-th worker after $l_i$ iterations. From the explicit form (4) of the remaining error of the executed iterative algorithm, we have

$$\mathbf{y}_i^{(l_i)} = \mathbf{y}_i^* + \epsilon_i^{(l_i)} = \mathbf{y}_i^* + \mathbf{B}^{l_i} \epsilon_i^{(0)}. \tag{49}$$

Therefore, from the definition $\mathbf{Y}^{(T_{\text{dl}})} = [\mathbf{y}_1^{(l_1)}, \mathbf{y}_2^{(l_2)}, \ldots, \mathbf{y}_n^{(l_n)}]$ (see Algorithm 1) and equation (48) and (49),

$$
\begin{aligned}
\mathbf{Y}^{(T_{\text{dl}})} &= [\mathbf{y}_1^{(l_1)}, \mathbf{y}_2^{(l_2)}, \ldots, \mathbf{y}_n^{(l_n)}] \\
&= [\mathbf{y}_1^*, \mathbf{y}_2^*, \ldots, \mathbf{y}_n^*] + [\epsilon_1^{(l_1)}, \epsilon_2^{(l_2)}, \ldots, \epsilon_n^{(l_n)}] \\
&= \mathbf{X}^* \cdot \mathbf{G} + [\mathbf{B}^{l_1} \epsilon_1^{(0)}, \ldots, \mathbf{B}^{l_n} \epsilon_n^{(0)}].
\end{aligned} \tag{50}
$$

Plugging in (7), we get the explicit form of $\mathbf{E} = \hat{\mathbf{X}}^\top - \mathbf{X}^*$:

$$
\begin{aligned}
\hat{\mathbf{X}}^\top &= (\mathbf{G}\boldsymbol{\Lambda}^{-1}\mathbf{G}^\top)^{-1}\mathbf{G}\boldsymbol{\Lambda}^{-1}(\mathbf{Y}^{(T_{\text{dl}})})^\top \\
&= (\mathbf{G}\boldsymbol{\Lambda}^{-1}\mathbf{G}^\top)^{-1}\mathbf{G}\boldsymbol{\Lambda}^{-1}\left[\mathbf{G}^\top(\mathbf{X}^*)^\top + [\mathbf{B}^{l_1}\epsilon_1^{(0)}, \ldots, \mathbf{B}^{l_n}\epsilon_n^{(0)}]^\top\right] \\
&= (\mathbf{X}^*)^\top + (\mathbf{G}\boldsymbol{\Lambda}^{-1}\mathbf{G}^\top)^{-1}\mathbf{G}\boldsymbol{\Lambda}^{-1}\left[\mathbf{B}^{l_1}\epsilon_1^{(0)}, \ldots, \mathbf{B}^{l_n}\epsilon_n^{(0)}\right]^\top.
\end{aligned} \tag{51}
$$

From (6), (48) and the definition $\epsilon_i^{(0)} = \mathbf{y}_i^{(0)} - \mathbf{y}_i^*$ and $\mathbf{e}_i^{(l)} = \mathbf{x}_i^{(0)} - \mathbf{x}_i^*$, we have

$$[\epsilon_1^{(0)}, \epsilon_2^{(0)}, \ldots, \epsilon_n^{(0)}] = [\mathbf{e}_1^{(0)}, \mathbf{e}_2^{(0)}, \ldots, \mathbf{e}_k^{(0)}] \cdot \mathbf{G}. \tag{52}$$

### 8.3.3 Vectorization of the error matrix E

From property 2 of Lemma 8.1, vectorization does not change the Frobenius norm, so we have

$$\mathbb{E}[\|\mathbf{E}\|^2 \,|\mathbf{l}] = \mathbb{E}[\|\text{vec}(\mathbf{E})\|^2 \,|\mathbf{l}] = \mathbb{E}\left[\text{trace}\left(\text{vec}(\mathbf{E})\text{vec}(\mathbf{E})^\top\right)|\mathbf{l}\right]. \tag{53}$$

Therefore, to prove the conclusion of this theorem, i.e., $\mathbb{E}[\|\mathbf{E}\|^2 \,|\mathbf{l}] \leq \sigma_{\max}(\mathbf{G}^\top\mathbf{G})\text{trace}\left[(\mathbf{G}\boldsymbol{\Lambda}^{-1}\mathbf{G}^\top)^{-1}\right]$, we only need to show

$$\mathbb{E}\left[\text{trace}\left(\text{vec}(\mathbf{E})\text{vec}(\mathbf{E})^\top\right)|\mathbf{l}\right] \leq \sigma_{\max}(\mathbf{G}^\top\mathbf{G})\text{trace}\left[(\mathbf{G}\boldsymbol{\Lambda}^{-1}\mathbf{G}^\top)^{-1}\right]. \tag{54}$$

### 8.3.4 Express the mean-squared error using the vectorization form

Now we prove (54). From (51), we have

$$\mathbf{E}^\top = (\mathbf{G}\boldsymbol{\Lambda}^{-1}\mathbf{G}^\top)^{-1}\mathbf{G}\boldsymbol{\Lambda}^{-1}[\mathbf{B}^{l_1}\epsilon_1^{(0)}, \ldots, \mathbf{B}^{l_n}\epsilon_n^{(0)}]^\top, \tag{55}$$

which is the same as

$$\mathbf{E} = [\mathbf{B}^{l_1}\epsilon_1^{(0)}, \ldots, \mathbf{B}^{l_n}\epsilon_n^{(0)}] \cdot [(\mathbf{G}\boldsymbol{\Lambda}^{-1}\mathbf{G}^\top)^{-1}\mathbf{G}\boldsymbol{\Lambda}^{-1}]^\top. \tag{56}$$

From property 1 of Lemma 8.1, (56) means

$$
\begin{aligned}
\text{vec}(\mathbf{E}) &= \left[(\mathbf{G}\boldsymbol{\Lambda}^{-1}\mathbf{G}^\top)^{-1}\mathbf{G}\boldsymbol{\Lambda}^{-1} \otimes \mathbf{I}_N\right] \cdot \text{vec}([\mathbf{B}^{l_1}\epsilon_1^{(0)}, \ldots, \mathbf{B}^{l_n}\epsilon_n^{(0)}]) \\
&= \left[(\mathbf{G}\boldsymbol{\Lambda}^{-1}\mathbf{G}^\top)^{-1}\mathbf{G}\boldsymbol{\Lambda}^{-1} \otimes \mathbf{I}_N\right] \cdot \text{diag}[\mathbf{B}^{l_1}, \ldots, \mathbf{B}^{l_n}] \cdot \text{vec}([\epsilon_1^{(0)}, \ldots, \epsilon_n^{(0)}]).
\end{aligned} \tag{57}
$$

Define

$$\mathbf{L} = (\mathbf{G}\boldsymbol{\Lambda}^{-1}\mathbf{G}^\top)^{-1}\mathbf{G}\boldsymbol{\Lambda}^{-1}, \tag{58}$$

$$\mathbf{D} = \text{diag}[\mathbf{B}^{l_1}, \ldots, \mathbf{B}^{l_n}], \tag{59}$$

and

$$\mathbf{E}_0 = \text{vec}([\epsilon_1^{(0)}, \ldots, \epsilon_n^{(0)}]). \tag{60}$$

Then,

$$\text{vec}(\mathbf{E}) = (\mathbf{L} \otimes \mathbf{I}_N) \cdot \mathbf{D} \cdot \mathbf{E}_0. \tag{61}$$

Therefore,

$$\mathbb{E}\left[\text{trace}\left(\text{vec}(\mathbf{E})\text{vec}(\mathbf{E})^\top\right)|\mathbf{l}\right] = \text{trace}\left((\mathbf{L} \otimes \mathbf{I}_N \cdot \mathbf{D})\mathbb{E}[\mathbf{E}_0\mathbf{E}_0^\top |\mathbf{l}](\mathbf{L} \otimes \mathbf{I}_N \cdot \mathbf{D})^\top\right). \tag{62}$$

### 8.3.5 Bounding the term $\mathbb{E}[\mathbf{E}_0\mathbf{E}_0^\top|\mathbf{l}]$ using the maximum eigenvalue $\sigma_{\mathbf{max}}(\mathbf{G}^\top\mathbf{G})$

Note that $\mathbf{E}_0 = \mathrm{vec}([\boldsymbol{\epsilon}_1^{(0)}, \ldots, \boldsymbol{\epsilon}_n^{(0)}])$. From (52), we have

$$[\boldsymbol{\epsilon}_1^{(0)}, \boldsymbol{\epsilon}_2^{(0)}, \ldots, \boldsymbol{\epsilon}_n^{(0)}] = [\mathbf{e}_1^{(0)}, \mathbf{e}_2^{(0)}, \ldots, \mathbf{e}_k^{(0)}] \cdot \mathbf{G}. \tag{63}$$

Therefore, using property 1 of Lemma 8.1, we have

$$\mathbf{E}_0 = (\mathbf{G}^\top \otimes \mathbf{I}_N) \cdot \mathrm{vec}([\mathbf{e}_1^{(0)}, \mathbf{e}_2^{(0)}, \ldots, \mathbf{e}_k^{(0)}]). \tag{64}$$

From Assumption 1, the covariance of $\mathbf{e}_i^{(0)}$ is

$$\mathbb{E}[\mathbf{e}_i^{(0)}(\mathbf{e}_i^{(0)})^\top|\mathbf{l}] = \mathbf{C}_E, i = 1, \ldots, k. \tag{65}$$

Therefore, from (64), we have

$$
\begin{aligned}
\mathbb{E}[\mathbf{E}_0\mathbf{E}_0^\top|\mathbf{l}] =& (\mathbf{G}^\top \otimes \mathbf{I}_N) \cdot \mathbb{E}[\mathrm{vec}([\mathbf{e}_1^{(0)}, \mathbf{e}_2^{(0)}, \ldots, \mathbf{e}_k^{(0)}]) \cdot \\
& \mathrm{vec}([\mathbf{e}_1^{(0)}, \mathbf{e}_2^{(0)}, \ldots, \mathbf{e}_k^{(0)}])^\top|\mathbf{l}] \cdot (\mathbf{G}^\top \otimes \mathbf{I}_N)^\top \\
\stackrel{(a)}{=}& (\mathbf{G}^\top \otimes \mathbf{I}_N) \cdot (\mathbf{I}_k \otimes \mathbf{C}_E) \cdot (\mathbf{G}^\top \otimes \mathbf{I}_N)^\top \\
=& (\mathbf{G}^\top \otimes \mathbf{I}_N) \cdot (\mathbf{I}_k \otimes \mathbf{C}_E) \cdot (\mathbf{G} \otimes \mathbf{I}_N) \\
\stackrel{(b)}{=}& (\mathbf{G}^\top \cdot \mathbf{I}_k \cdot \mathbf{G}) \otimes (\mathbf{I}_N \cdot \mathbf{C}_E \cdot \mathbf{I}_N) \\
=& \mathbf{G}^\top\mathbf{G} \otimes \mathbf{C}_E \\
\stackrel{(c)}{\preceq}& \sigma_{\mathrm{max}}(\mathbf{G}^\top\mathbf{G})\mathbf{I}_n \otimes \mathbf{C}_E,
\end{aligned}
\tag{66}
$$

where (a) is from (65), (b) and (c) follow respectively from property 3 and property 5 of Lemma 8.1.

If $\mathbf{G}$ has orthonormal rows, the eigenvalues of $\mathbf{G}^\top\mathbf{G}$ (which is an $n \times n$ matrix) are all in $(0, 1]$. This is why we can remove the term $\sigma_{\mathrm{max}}(\mathbf{G}^\top\mathbf{G})$ in (11) when $\mathbf{G}$ has orthonormal rows. In what follows, we assume $\mathbf{G}$ has orthonormal rows, and the result when $\mathbf{G}$ does not have orthonormal rows follows naturally.

Assuming $\mathbf{G}$ has orthonormal rows, we have

$$\mathbb{E}[\mathbf{E}_0\mathbf{E}_0^\top|\mathbf{l}] \preceq \mathbf{I}_n \otimes \mathbf{C}_E. \tag{67}$$

Plugging (67) into (62), we have

$$\mathbb{E}\left[\mathrm{trace}\left(\mathrm{vec}(\mathbf{E})\mathrm{vec}(\mathbf{E})^\top\right)|\mathbf{l}\right] \leq \mathrm{trace}\left((\mathbf{L} \otimes \mathbf{I}_N) \cdot \mathbf{D}(\mathbf{I}_n \otimes \mathbf{C}_E)\mathbf{D}^\top(\mathbf{L} \otimes \mathbf{I}_N)^\top\right), \tag{68}$$

where $\mathbf{D} = \mathrm{diag}[\mathbf{B}^{l_1}, \ldots, \mathbf{B}^{l_n}]$. Therefore,

$$\mathbf{D}(\mathbf{I}_n \otimes \mathbf{C}_E)\mathbf{D}^\top = \mathrm{diag}[\mathbf{B}^{l_1}\mathbf{C}_E(\mathbf{B}^\top)^{l_1}, \ldots, \mathbf{B}^{l_n}\mathbf{C}_E(\mathbf{B}^\top)^{l_n}]. \tag{69}$$

From the definition of $\mathbf{C}(l_i)$ in (9),

$$\mathbf{D}(\mathbf{I}_n \otimes \mathbf{C}_E)\mathbf{D}^\top = \mathrm{diag}[\mathbf{C}(l_1), \ldots, \mathbf{C}(l_n)]. \tag{70}$$

### 8.3.6 Reducing the dimensionality of $\mathbf{D}(\mathbf{I}_n \otimes \mathbf{C}_E)\mathbf{D}^\top$ in the trace expression using property 7 in Lemma 8.1

From Property 7 in Lemma 8.1, we can simplify (71):

$$
\begin{aligned}
\mathbb{E}\left[\mathrm{trace}\left(\mathrm{vec}(\mathbf{E})\mathrm{vec}(\mathbf{E})^\top\right)|\mathbf{l}\right] \leq& \mathrm{trace}\left((\mathbf{L} \otimes \mathbf{I}_N) \cdot \mathbf{D}(\mathbf{I}_n \otimes \mathbf{C}_E)\mathbf{D}^\top(\mathbf{L} \otimes \mathbf{I}_N)^\top\right) \\
\stackrel{(a)}{=}& \mathrm{trace}\left((\mathbf{L} \otimes \mathbf{I}_N) \cdot \mathrm{diag}[\mathbf{C}(l_1), \ldots, \mathbf{C}(l_n)](\mathbf{L} \otimes \mathbf{I}_N)^\top\right) \\
\stackrel{(b)}{=}& \mathrm{trace}\left[\mathbf{L} \cdot \mathrm{diag}[\mathrm{trace}(\mathbf{C}(l_1)), \ldots, \mathrm{trace}(\mathbf{C}(l_1))]\mathbf{L}^\top\right] \\
\stackrel{(c)}{=}& \mathrm{trace}\left(\mathbf{L}\boldsymbol{\Lambda}\mathbf{L}^\top\right),
\end{aligned}
\tag{71}
$$

where (a) is from (70), (b) is from Property 7 and (c) is from the definition of $\mathbf{\Lambda}$ in (8). Equation (71) can be further simplified to

$$\mathbb{E}\left[\mathrm{trace}\left(\mathrm{vec}(\mathbf{E})\mathrm{vec}(\mathbf{E})^\top\right)|\mathbf{l}\right] \leq \mathrm{trace}\left(\mathbf{L}\mathbf{\Lambda}\mathbf{L}^\top\right)$$

$$\overset{(a)}{=} \mathrm{trace}\left((\mathbf{G}\mathbf{\Lambda}^{-1}\mathbf{G}^\top)^{-1}\mathbf{G}\mathbf{\Lambda}^{-1}\mathbf{\Lambda}((\mathbf{G}\mathbf{\Lambda}^{-1}\mathbf{G}^\top)^{-1}\mathbf{G}\mathbf{\Lambda}^{-1})^\top\right) \qquad (72)$$

$$= \mathrm{trace}((\mathbf{G}\mathbf{\Lambda}^{-1}\mathbf{G}^\top)^{-1}),$$

where (a) is from the definition of the decoding matrix $\mathbf{L} = (\mathbf{G}\mathbf{\Lambda}^{-1}\mathbf{G}^\top)^{-1}\mathbf{G}\mathbf{\Lambda}^{-1}$. Thus, we have completed the proof of Theorem 3.1 for the case when $\mathbf{G}$ has orthonormal rows. As we argued earlier, the proof when $\mathbf{G}$ does not have orthonormal rows follows immediately (see the text after (66)).

## 8.4 Proof of Theorem 4.2

First, we need the following corollary of Theorem 3.1.

*Corollary* 8.1. Suppose the i.i.d. Assumption 1 holds and the matrix $\mathbf{G}_{k\times n}$ is a submatrix of an $n \times n$ orthonormal matrix. Assume that the symmetric matrix $\mathbf{F}\mathbf{\Lambda}\mathbf{F}^\top$ has the block form

$$\mathbf{F}\mathbf{\Lambda}\mathbf{F}^\top = \begin{bmatrix} \mathbf{J}_1 & \mathbf{J}_2 \\ \mathbf{J}_2^\top & \mathbf{J}_4 \end{bmatrix}_{n\times n}, \qquad (73)$$

that is, $(\mathbf{J}_1)_{k\times k}$ is $\mathbf{G}\mathbf{\Lambda}\mathbf{G}^\top$, $(\mathbf{J}_2)_{k\times(n-k)}$ is $\mathbf{G}\mathbf{\Lambda}\mathbf{H}^\top$, and $(\mathbf{J}_4)_{(n-k)\times(n-k)}$ is $\mathbf{H}\mathbf{\Lambda}\mathbf{H}^\top$. Then, we have

$$\mathbb{E}[\|\mathbf{E}\|^2|\mathbf{l}] \leq \mathrm{trace}(\mathbf{J}_1) - \mathrm{trace}(\mathbf{J}_2\mathbf{J}_4^{-1}\mathbf{J}_2^\top). \qquad (74)$$

*Proof.* First, note that

$$\mathbf{G} = [\mathbf{I}_k, \mathbf{0}_{k,n-k}]\,\mathbf{F}. \qquad (75)$$

Therefore,

$$\mathbf{G}\mathbf{\Lambda}^{-1}\mathbf{G}^\top = [\mathbf{I}_k, \mathbf{0}_{k,n-k}]\,\mathbf{F}\mathbf{\Lambda}^{-1}\mathbf{F}^\top\,[\mathbf{I}_k, \mathbf{0}_{k,n-k}]^\top$$

$$\overset{(a)}{=} [\mathbf{I}_k, \mathbf{0}_{k,n-k}]\,(\mathbf{F}\mathbf{\Lambda}\mathbf{F}^\top)^{-1}\,[\mathbf{I}_k, \mathbf{0}_{k,n-k}]^\top, \qquad (76)$$

where (a) is from $\mathbf{F}^\top\mathbf{F} = \mathbf{I}_n$. Now take the inverse of both sides of (73), we have

$$(\mathbf{F}\mathbf{\Lambda}\mathbf{F}^\top)^{-1} = \begin{bmatrix} (\mathbf{J}_1 - \mathbf{J}_2\mathbf{J}_4^{-1}\mathbf{J}_2^\top)^{-1} & * \\ * & * \end{bmatrix}_{n\times n}, \qquad (77)$$

where $*$ is used as a substitute for matrices that are unimportant for our argument. Thus, comparing (76) and (77),

$$\mathbf{G}\mathbf{\Lambda}^{-1}\mathbf{G}^\top = (\mathbf{J}_1 - \mathbf{J}_2\mathbf{J}_4^{-1}\mathbf{J}_2^\top)^{-1}, \qquad (78)$$

which means

$$(\mathbf{G}\mathbf{\Lambda}^{-1}\mathbf{G}^\top)^{-1} = \mathbf{J}_1 - \mathbf{J}_2\mathbf{J}_4^{-1}\mathbf{J}_2^\top. \qquad (79)$$

From (11) and (79), the theorem follows. □

From Corollary 8.1, for fixed $l_i, 1 \leq i \leq n$,

$$\mathbb{E}[\|\mathbf{E}_{\mathrm{coded}}\|^2|\mathbf{l}] \leq \mathrm{trace}(\mathbf{J}_1) - \mathrm{trace}(\mathbf{J}_2\mathbf{J}_4^{-1}\mathbf{J}_2^\top). \qquad (80)$$

We will show that

$$\mathbb{E}_f[\mathrm{trace}(\mathbf{J}_1)] = \mathbb{E}_f\left[\|\mathbf{E}_{\mathrm{uncoded}}\|^2\right], \qquad (81)$$

which completes the proof. To show (81), first note that from (18),

$$\mathbb{E}_f\left[\|\mathbf{E}_{\mathrm{uncoded}}\|^2\right] = k\mathbb{E}_f[\mathrm{trace}(\mathbf{C}(l_1))]. \qquad (82)$$

Since $\mathbf{G} := [g_{j,i}]$ is a submatrix of a Fourier matrix, we have $|g_{ji}|^2 = 1/n$. Thus, $\mathbf{J}_1 = \mathbf{G}\mathbf{\Lambda}\mathbf{G}^\top$ satisfies

$$\mathrm{trace}(\mathbf{J}_1) = \sum_{j=1}^{k}\sum_{i=1}^{n}|g_{ji}|^2\mathrm{trace}(\mathbf{C}(l_i)) = \frac{k}{n}\sum_{i=1}^{n}\mathrm{trace}(\mathbf{C}(l_i)).$$

Therefore,

$$\mathbb{E}_f[\mathrm{trace}(\mathbf{J}_1)] = k\mathbb{E}_f[\mathrm{trace}(\mathbf{C}(l_1))]. \qquad (83)$$

which, along with (82), completes the proof of (81), and hence also the proof of Theorem 4.2.

## 8.5 Proof of Theorem 8.1

The proof follows the same procedure as the proof of Theorem 3.1. Basically, we can obtain exactly the same results from (45) to (64) except that all $\mathbf{\Lambda}$ are replaced with $\tilde{\mathbf{\Lambda}}$. However, now that we assume the solutions of the linear inverse problems satisfy 4, we have

$$\mathbb{E}[\text{vec}([\mathbf{e}_1^{(0)}, \mathbf{e}_2^{(0)}, \ldots, \mathbf{e}_k^{(0)}])\text{vec}([\mathbf{e}_1^{(0)}, \mathbf{e}_2^{(0)}, \ldots, \mathbf{e}_k^{(0)}])^\top | \mathbf{l}] = \mathbf{I}_k \otimes \mathbf{C}_E + (\mathbf{1}_k \mathbf{1}_k^\top) \otimes \mathbf{C}_{\text{cor}}. \quad (84)$$

Note that the first part $\mathbf{I}_k \otimes \mathbf{C}_E$ is exactly the same as in the proof of Theorem 3.1, so all conclusions until (71) can still be obtained (note that $\sigma_{\max}(\mathbf{G}^\top \mathbf{G})$ should be added in the general case) for this part. More specifically, this means (71) can be modified to

$$\mathbb{E}\left[\text{trace}\left(\text{vec}(\mathbf{E})\text{vec}(\mathbf{E})^\top\right) | \mathbf{l}\right] \leq \text{trace}\left((\mathbf{L} \otimes \mathbf{I}_N) \cdot \mathbf{D}\mathbf{\Sigma}\mathbf{D}^\top(\mathbf{L} \otimes \mathbf{I}_N)^\top\right)$$
$$+ \sigma_{\max}(\mathbf{G}^\top \mathbf{G})\text{trace}\left(\mathbf{L}\mathbf{\Lambda}\mathbf{L}^\top\right), \quad (85)$$

where the second term $\sigma_{\max}(\mathbf{G}^\top \mathbf{G})\text{trace}\left(\mathbf{L}\mathbf{\Lambda}\mathbf{L}^\top\right)$ is the same as in (71) because of the first part $\mathbf{I}_k \otimes \mathbf{C}_E$ in (84). However, the first term $\text{trace}\left((\mathbf{L} \otimes \mathbf{I}_N) \cdot \mathbf{D}\mathbf{\Sigma}\mathbf{D}^\top(\mathbf{L} \otimes \mathbf{I}_N)^\top\right)$ is from the correlation between different inputs, and the matrix $\mathbf{\Sigma}$ is

$$\mathbf{\Sigma} = (\mathbf{G}^\top \otimes \mathbf{I}_N) \cdot ((\mathbf{1}_k \mathbf{1}_k^\top) \otimes \mathbf{C}_{\text{cor}}) \cdot (\mathbf{G}^\top \otimes \mathbf{I}_N)^\top, \quad (86)$$

which is obtained by adding the second term $(\mathbf{1}_k \mathbf{1}_k^\top) \otimes \mathbf{C}_{\text{cor}}$ in (84) into the step (a) in (66). From Property 6 of Lemma 8.1, (86) can be simplified to

$$\mathbf{\Sigma} = (\mathbf{G}^\top \mathbf{1}_k \mathbf{1}_k^\top \mathbf{G}) \otimes \mathbf{C}_{\text{cor}}. \quad (87)$$

Therefore, from the definition of (59)

$$\mathbf{D}\mathbf{\Sigma}\mathbf{D}^\top = \text{diag}[\mathbf{B}^{l_1}, \ldots, \mathbf{B}^{l_n}] \cdot (\mathbf{G}^\top \mathbf{1}_k \mathbf{1}_k^\top \mathbf{G}) \otimes \mathbf{C}_{\text{cor}} \cdot \text{diag}[(\mathbf{B}^\top)^{l_1}, \ldots, (\mathbf{B}^\top)^{l_n}]. \quad (88)$$

Define the column vector $\mathbf{h} = \mathbf{G}^\top \mathbf{1}_k := [h_1, h_2, \ldots h_n]^\top$. Then, $(\mathbf{G}^\top \mathbf{1}_k \mathbf{1}_k^\top \mathbf{G}) \otimes \mathbf{C}_{\text{cor}}$ can be written as a block matrix where the block on the $i$-th row and the $j$-th column is $h_i h_j^* \mathbf{C}_{\text{cor}}$. Therefore, After left-multiplying the block diagonal matrix $\text{diag}[\mathbf{B}^{l_1}, \ldots, \mathbf{B}^{l_n}]$ and right-multiplying $\text{diag}[(\mathbf{B}^\top)^{l_1}, \ldots, (\mathbf{B}^\top)^{l_n}]$, we obtain

$$\mathbf{D}\mathbf{\Sigma}\mathbf{D}^\top = \tilde{\mathbf{\Psi}} = [\tilde{\mathbf{\Psi}}_{i,j}], \quad (89)$$

where the block $\tilde{\mathbf{\Psi}}_{i,j}$ on the $i$-th row and the $j$-th column is $h_i h_j^* \mathbf{B}^{l_i} \mathbf{C}_{\text{cor}} (\mathbf{B}^\top)^{l_j}$. From Property 7 of Lemma 8.1, we have

$$\text{trace}[(\mathbf{L} \otimes \mathbf{I}_N) \cdot \mathbf{D}\mathbf{\Sigma}\mathbf{D}^\top(\mathbf{L} \otimes \mathbf{I}_N)^\top] \overset{(a)}{=} \text{trace}\left[\mathbf{L}\left[\text{trace}[\tilde{\mathbf{\Psi}}_{i,j}]\right]\mathbf{L}^\top\right]$$
$$\overset{(b)}{=} \text{trace}\left[\mathbf{L}\left[h_i h_j^* \text{trace}[\mathbf{B}^{l_i} \mathbf{C}_{\text{cor}}(\mathbf{B}^\top)^{l_j}]\right]\mathbf{L}^\top\right]$$
$$\overset{(c)}{=} \text{trace}\left[\mathbf{L}\text{diag}(\mathbf{h})\left[\text{trace}[\mathbf{B}^{l_i} \mathbf{C}_{\text{cor}}(\mathbf{B}^\top)^{l_j}]\right]\text{diag}(\mathbf{h}^\top)\mathbf{L}^\top\right]$$
$$\overset{(d)}{=} \text{trace}\left[\mathbf{L}\text{diag}\{\mathbf{G}^\top \mathbf{1}_k\} \cdot \mathbf{\Psi} \cdot \text{diag}\{\mathbf{G}^\top \mathbf{1}_k\}^\top \mathbf{L}^\top\right], \quad (90)$$

where step (a) is from Property 7 of Lemma 8.1 and the notation $\left[\text{trace}[\tilde{\mathbf{\Psi}}_{i,j}]\right]$ means the $n \times n$ matrix with entries $\text{trace}[\tilde{\mathbf{\Psi}}_{i,j}]$, (b) is from the definition of $\tilde{\mathbf{\Psi}}_{i,j}$ below (90), (c) is from the definition $\mathbf{h} = \mathbf{G}^\top \mathbf{1}_k := [h_1, h_2, \ldots h_n]^\top$, and (d) is from the definition of the matrix $\mathbf{\Psi}$ in (35). Plugging (90) into (85), we obtain

$$\mathbb{E}\left[\text{trace}\left(\text{vec}(\mathbf{E})\text{vec}(\mathbf{E})^\top\right) | \mathbf{l}\right] \leq \text{trace}\left[\mathbf{L}\text{diag}\{\mathbf{G}^\top \mathbf{1}_k\} \cdot \mathbf{\Psi} \cdot \text{diag}\{\mathbf{G}^\top \mathbf{1}_k\}^\top \mathbf{L}^\top\right]$$
$$+ \sigma_{\max}(\mathbf{G}^\top \mathbf{G})\text{trace}\left(\mathbf{L}\mathbf{\Lambda}\mathbf{L}^\top\right) \quad (91)$$
$$= \text{trace}[\mathbf{L}\tilde{\mathbf{\Lambda}}\mathbf{L}^\top],$$

where $\tilde{\mathbf{\Lambda}} = \sigma_{\max}(\mathbf{G}^\top \mathbf{G})\mathbf{\Lambda} + \text{diag}\{\mathbf{G}^\top \mathbf{1}_k\} \cdot \mathbf{\Psi} \cdot \text{diag}\{\mathbf{G}^\top \mathbf{1}_k\}^\top$, which is the same as in (34). Therefore

$$\mathbb{E}\left[\text{trace}\left(\text{vec}(\mathbf{E})\text{vec}(\mathbf{E})^\top\right) | \mathbf{l}\right] \leq \text{trace}\left(\mathbf{L}\tilde{\mathbf{\Lambda}}\mathbf{L}^\top\right)$$
$$\overset{(a)}{=} \text{trace}\left((\mathbf{G}\tilde{\mathbf{\Lambda}}^{-1}\mathbf{G}^\top)^{-1}\mathbf{G}\tilde{\mathbf{\Lambda}}^{-1}\tilde{\mathbf{\Lambda}}((\mathbf{G}\tilde{\mathbf{\Lambda}}^{-1}\mathbf{G}^\top)^{-1}\mathbf{G}\tilde{\mathbf{\Lambda}}^{-1})^\top\right) \quad (92)$$
$$= \text{trace}((\mathbf{G}\tilde{\mathbf{\Lambda}}^{-1}\mathbf{G}^\top)^{-1}),$$

where (a) is from the definition of the decoding matrix $\mathbf{L} = (\mathbf{G}\tilde{\mathbf{\Lambda}}^{-1}\mathbf{G}^\top)^{-1}\mathbf{G}\tilde{\mathbf{\Lambda}}^{-1}$.

## 8.6 Proof of Theorem 4.1

In this section, we compute the residual error of the uncoded linear inverse algorithm. From (4), in the uncoded scheme, the overall error is

$$
\begin{aligned}
\mathbb{E}\left[\|\mathbf{E}_{\text{uncoded}}\|^2 \,|\mathbf{l}\right] &= \mathbb{E}\left[\left\|[\mathbf{e}_1^{(l_1)}, \mathbf{e}_2^{(l_2)} \ldots, \mathbf{e}_k^{(l_k)}]\right\|^2 |\mathbf{l}\right] \\
&= \sum_{i=1}^{k} \mathbb{E}\left[\left\|[\mathbf{e}_i^{(l_i)}]\right\|^2 |\mathbf{l}\right] \\
&= \sum_{i=1}^{k} \text{trace}\left(\mathbb{E}\left[\mathbf{e}_i^{(l_i)}(\mathbf{e}_i^{(l_i)})^\top |\mathbf{l}\right]\right) \\
&\stackrel{(a)}{=} \sum_{i=1}^{k} \text{trace}\left(\mathbb{E}\left[\mathbf{B}^{l_i}\mathbf{e}_i^{(0)}(\mathbf{B}^{l_i}\mathbf{e}_i^{(0)})^\top |\mathbf{l}\right]\right) \\
&= \sum_{i=1}^{k} \text{trace}\left(\mathbf{B}^{l_i}\mathbb{E}\left[\mathbf{e}_i^{(0)}(\mathbf{e}_i^{(0)})^\top |\mathbf{l}\right](\mathbf{B}^{l_i})^\top\right) \\
&= \sum_{i=1}^{k} \text{trace}\left(\mathbf{B}^{l_i} \cdot \mathbf{C}_E \cdot (\mathbf{B}^{l_i})^\top\right) \\
&= \sum_{i=1}^{k} \text{trace}\left(\mathbf{B}^{l_i}\mathbf{C}_E(\mathbf{B}^{l_i})^\top\right) \\
&\stackrel{(b)}{=} \sum_{i=1}^{k} \text{trace}\left(\mathbf{C}(l_i)\right),
\end{aligned}
\tag{93}
$$

where (a) is from (4) and (b) is from the definition of $\mathbf{C}(l_i)$ in (9). Thus, we have computed the closed-form of the MSE of the uncoded method. To prove (18), we note that from the i.i.d. assumption of $l_i$,

$$
\mathbb{E}_f\left[\|\mathbf{E}_{\text{uncoded}}\|^2\right] = \mathbb{E}_f\left[\sum_{i=1}^{k} \text{trace}\left(\mathbf{C}(l_i)\right)\right] = k\mathbb{E}_f[\text{trace}(\mathbf{C}(l_1))].
\tag{94}
$$

## 8.7 Proof of Theorem 4.4

From Theorem 4.2,

$$
\mathbb{E}_f\left[\|\mathbf{E}_{\text{uncoded}}\|^2\right] - \mathbb{E}_f\left[\|\mathbf{E}_{\text{coded}}\|^2\right] \geq \mathbb{E}_f[\text{trace}(\mathbf{J}_2\mathbf{J}_4^{-1}\mathbf{J}_2^\top)].
\tag{95}
$$

We now argue that to show (22), we only need to show

$$
\lim_{n\to\infty} \frac{1}{n-k}\mathbb{E}_f[\text{trace}(\mathbf{J}_2\mathbf{J}_4^{-1}\mathbf{J}_2^\top)] \geq \frac{\text{var}_f[\text{trace}(\mathbf{C}(l_1))]}{\mathbb{E}_f[\text{trace}(\mathbf{C}(l_1))]},
\tag{96}
$$

because then, we have

$$
\begin{aligned}
\lim_{n\to\infty} \frac{1}{(n-k)}\left[\mathbb{E}_f\left[\|\mathbf{E}_{\text{uncoded}}\|^2\right] - \mathbb{E}_f\left[\|\mathbf{E}_{\text{coded}}\|^2\right]\right] &\stackrel{(a)}{\geq} \lim_{n\to\infty} \frac{1}{(n-k)}\mathbb{E}_f[\text{trace}(\mathbf{J}_2\mathbf{J}_4^{-1}\mathbf{J}_2^\top)] \\
&\stackrel{(b)}{\geq} \frac{\text{var}_f[\text{trace}(\mathbf{C}(l_1))]}{\mathbb{E}_f[\text{trace}(\mathbf{C}(l_1))]},
\end{aligned}
\tag{97}
$$

where (a) follows from (95) and (b) follows from (96).

Also note that after we prove (22), then using (20), we have

$$
\mathbb{E}_f\left[\|\mathbf{E}_{\text{uncoded}}\|^2\right] - \mathbb{E}_f\left[\|\mathbf{E}_{\text{rep}}\|^2\right] \leq (n-k)\mathbb{E}_f[\text{trace}(\mathbf{C}(l_1))],
\tag{98}
$$

so we have

$$
\lim_{n\to\infty} \frac{1}{(n-k)} \left[ \mathbb{E}_f\left[ \left\| \mathbf{E}_{\text{uncoded}} \right\|^2 \right] - \mathbb{E}_f\left[ \left\| \mathbf{E}_{\text{rep}} \right\|^2 \right] \right]
$$

$$
\leq \mathbb{E}_f[\text{trace}(\mathbf{C}(l_1))]
$$

$$
\overset{(a)}{\leq} \frac{1}{\rho} \frac{\text{var}_f[\text{trace}(\mathbf{C}(l_1))]}{\mathbb{E}_f[\text{trace}(\mathbf{C}(l_1))]} \tag{99}
$$

$$
\leq \frac{1}{\rho} \lim_{n\to\infty} \frac{1}{(n-k)} \left[ \mathbb{E}_f\left[ \left\| \mathbf{E}_{\text{uncoded}} \right\|^2 \right] - \mathbb{E}_f\left[ \left\| \mathbf{E}_{\text{coded}} \right\|^2 \right] \right],
$$

which means coded computation beats uncoded computation. Note that step (a) holds because of the variance heavy-tail property.

Therefore, we only need to prove (96). The proof of (96) is divided into two steps, and intuition behind each step is provided along the proof. The main intuition is that the Fourier structure of the matrix $\mathbf{F}$ makes the matrix $\mathbf{J}_4$ concentrates around its mean value, which makes the most tricky term $\mathbb{E}_f[\text{trace}(\mathbf{J}_2 \mathbf{J}_4^{-1} \mathbf{J}_2^\top)]$ analyzable.

### 8.7.1 Exploiting the Fourier structure to obtain a Toeplitz covariance matrix

First, we claim that when $\mathbf{F}_{n\times n}$ is the Fourier transform matrix, the matrix $\mathbf{F}\mathbf{\Lambda}\mathbf{F}^\top$ in (73)

$$
\mathbf{F}\mathbf{\Lambda}\mathbf{F}^\top = \begin{bmatrix} \mathbf{J}_1 & \mathbf{J}_2 \\ \mathbf{J}_2^\top & \mathbf{J}_4 \end{bmatrix}_{n\times n}, \tag{100}
$$

is a Toeplitz matrix composed of the Fourier coefficients of the sequence (vector) $s = [\text{trace}(\mathbf{C}(l_1)), \ldots, \text{trace}(\mathbf{C}(l_n))]$. In what follows, we use the simplified notation

$$
s_j := \text{trace}(\mathbf{C}(l_{j+1})), j = 0, 1, \ldots, n - 1. \tag{101}
$$

*Lemma* 8.2. If

$$
\mathbf{F} = \left( \frac{w^{pq}}{\sqrt{n}} \right)_{p,q=0,1,\ldots,n-1}, \tag{102}
$$

where $w = \exp(-2\pi i/n)$, then

$$
\mathbf{F}\mathbf{\Lambda}\mathbf{F}^\top = \text{Toeplitz}[\tilde{s}_p]_{p=0,1,\ldots,n-1}, \tag{103}
$$

where

$$
\tilde{s}_p = \frac{1}{n} \sum_{j=0}^{n-1} w^{-pj} s_j \tag{104}
$$

*Proof.* The entry on the $l$-th row and the $m$-th column of $\mathbf{F}\mathbf{\Lambda}\mathbf{F}^\top$ is

$$
[\mathbf{F}\mathbf{\Lambda}\mathbf{F}^\top]_{l,m} = \sum_{j=0}^{n-1} \frac{w^{lj}}{\sqrt{n}} \frac{w^{-mj}}{\sqrt{n}} s_j = \frac{1}{n} \sum_{j=0}^{n-1} w^{(l-m)j} s_j. \tag{105}
$$

Thus, Lemma 8.2 holds. □

Therefore, the variance of all entries of $\mathbf{F}\mathbf{\Lambda}\mathbf{F}^\top$ is the same because

$$
\text{var}_f[\tilde{s}_p] = \text{var}_f \left[ \frac{1}{n} \sum_{j=0}^{n-1} w^{-pj} s_j \right] = \frac{1}{n} \text{var}_f[s_0] =: \frac{1}{n} v. \tag{106}
$$

Further, the means of all diagonal entries of $\mathbf{F}\mathbf{\Lambda}\mathbf{F}^\top$ are

$$
\mathbb{E}_f[\tilde{s}_0] = \mathbb{E}_f[s_0] =: \mu, \tag{107}
$$

while the means of all off-diagonal entries are

$$
\mathbb{E}_f[\tilde{s}_p] = \frac{1}{n} \sum_{j=0}^{n-1} w^{-pj} \mathbb{E}_f[s_j] = 0, \forall p \neq 0. \tag{108}
$$

### 8.7.2 Using the concentration of $\mathbf{J}_4$ to obtain the error when $n \to \infty$

From an intuitive perspective, when $n \to \infty$, the submatrix $\mathbf{J}_4$ concentrates at $\mu \mathbf{I}_{n-k}$ (see the above computation on the mean and variance of all entries). In this case

$$
\begin{aligned}
\mathbb{E}_f[\text{trace}(\mathbf{J}_2 \mathbf{J}_4^{-1} \mathbf{J}_2^\top)] &\approx \frac{1}{\mu} \mathbb{E}_f[\text{trace}(\mathbf{J}_2 \mathbf{J}_2^\top)] \\
&= \frac{1}{\mu} k(n-k) \text{var}[\tilde{s}_p] \\
&= \frac{n-k}{\mu} v \cdot \frac{k}{n}.
\end{aligned}
\tag{109}
$$

Therefore, we have

$$
\lim_{n \to \infty} \frac{1}{n-k} \mathbb{E}_f[\text{trace}(\mathbf{J}_2 \mathbf{J}_4^{-1} \mathbf{J}_2^\top)] = \frac{v}{\mu} = \frac{\text{var}_f[s_0]}{\mathbb{E}_f[s_0]}.
\tag{110}
$$

Now, we formalize the above intuitive statement. In fact, we will show a even stronger bound than the bound on the expected error.

*Lemma 8.3.* When $n - k = o(\sqrt{n})$, with high probability (in $1 - \mathcal{O}(\frac{(n-k)^2}{n})$),

$$
\frac{1}{n-k} \text{trace}(\mathbf{J}_2 \mathbf{J}_4^{-1} \mathbf{J}_2^\top) \geq \frac{1}{\mu + \epsilon} \left( \frac{k}{n} v - \epsilon \right),
\tag{111}
$$

for any $\epsilon > 0$.

After we prove Lemma 8.3, we obtain a bound on expectation using the fact that

$$
\frac{1}{n-k} \mathbb{E}_f[\text{trace}(\mathbf{J}_2 \mathbf{J}_4^{-1} \mathbf{J}_2^\top)] \geq (1 - \mathcal{O}(\frac{(n-k)^2}{n})) \frac{1}{\mu + \epsilon} \left( \frac{k}{n} v - \epsilon \right).
\tag{112}
$$

Thus, when $n \to \infty$ and $n - k = o(\sqrt{n})$,

$$
\lim_{n \to \infty} \frac{1}{n-k} \mathbb{E}_f[\text{trace}(\mathbf{J}_2 \mathbf{J}_4^{-1} \mathbf{J}_2^\top)] \geq \frac{v - \epsilon}{\mu + \epsilon} = \frac{\text{var}_f[s_0] - \epsilon}{\mathbb{E}_f[s_0] + \epsilon},
\tag{113}
$$

for all $\epsilon > 0$, which completes the proof of Theorem 4.4.

The proof of Lemma 8.3 relies on the concentration of $\text{trace}(\mathbf{J}_2 \mathbf{J}_2^\top)$ and the concentration of $\mathbf{J}_4$. In particular, when we prove the concentration of $\mathbf{J}_4$, we use the Gershgorin circle theorem [27]. First, we show the following Lemma.

*Lemma 8.4.* When $n - k = o(n)$, with high probability (in $1 - \mathcal{O}(\frac{n-k}{n})$)

$$
\frac{1}{n-k} \text{trace}(\mathbf{J}_2 \mathbf{J}_2^\top) \geq \frac{k}{n} v - \epsilon.
\tag{114}
$$

*Proof.* Since $(\mathbf{J}_2)_{k \times (n-k)} := [\mathbf{J}_{i,j}]$ ($\mathbf{J}_{i,j}$ represents the entry on the $i$-th row and the $j$-th column) is the upper-right submatrix of $\mathbf{F} \mathbf{\Lambda} \mathbf{F}^\top = \text{Toeplitz}[\tilde{s}_p]_{p=0,1,\dots,n-1}$,

$$
\text{trace}(\mathbf{J}_2 \mathbf{J}_2^\top) = \sum_{i=1}^{k} \sum_{j=1}^{n-k} |\mathbf{J}_{i,j}|^2 = \sum_{l=1}^{k} \sum_{m=k+1}^{n} |\tilde{s}_{m-l}|^2.
\tag{115}
$$

Since all entries in $\mathbf{J}_2$ have zero mean (because $l \neq m$ ever in (115) and from (108) all off-diagonal entries have zero mean) and have the same variance $\frac{v}{n}$ (see (106)),

$$
\mathbb{E}_f\left[ \frac{1}{n-k} \text{trace}(\mathbf{J}_2 \mathbf{J}_2^\top) \right] = \frac{1}{n-k} \cdot k(n-k) \mathbb{E}_f[|\tilde{s}_1|^2] \overset{(a)}{=} \frac{1}{n-k} \cdot k(n-k) \text{var}_f[\tilde{s}_1] = \frac{k}{n} v,
\tag{116}
$$

where (a) holds because $\mathbb{E}_f[\tilde{s}_1] = 0$. To prove (114), we compute the variance of $\text{trace}(\mathbf{J}_2 \mathbf{J}_2^\top)$ and use Chebyshev's inequality to bound the tail probability. Define

$$
\mu_B := \mathbb{E}_f[\text{trace}(\mathbf{J}_2 \mathbf{J}_2^\top)] \overset{(a)}{=} \frac{k(n-k)}{n} v,
\tag{117}
$$

where (a) follows from (116). From (115), we have

$$\text{trace}(\mathbf{J}_2\mathbf{J}_2^\top) \leq (n-k)\sum_{p=1}^{n-1}|\tilde{s}_p|^2 \overset{(a)}{=} (n-k)\left(\frac{1}{n}\sum_{j=0}^{n-1}s_j^2 - |\tilde{s}_0|^2\right), \tag{118}$$

where the last equality (a) holds due to Parseval's equality for the Fourier transform, which states that $\frac{1}{n}\sum_{j=0}^{n-1}s_j^2 = \sum_{p=0}^{n-1}|\tilde{s}_p|^2$. Then,

$$
\begin{aligned}
\text{var}_f\left[\frac{1}{n-k}\text{trace}(\mathbf{J}_2\mathbf{J}_2^\top)\right] &= \mathbb{E}_f\left[\left(\frac{1}{n-k}\text{trace}(\mathbf{J}_2\mathbf{J}_2^\top)\right)^2\right] - \mathbb{E}_f^2\left[\frac{1}{n-k}\text{trace}(\mathbf{J}_2\mathbf{J}_2^\top)\right] \\
&\overset{(a)}{\leq} \mathbb{E}_f\left[\left(\frac{1}{n}\sum_{j=0}^{n-1}s_j^2 - |\tilde{s}_0|^2\right)^2\right] - \frac{k^2}{n^2}v^2 \\
&\overset{(b)}{=} \mathbb{E}_f\left[\left(\frac{1}{n}\sum_{j=0}^{n-1}s_j^2 - (\frac{1}{n}\sum_{j=0}^{n-1}s_j)^2\right)^2\right] - \frac{k^2}{n^2}v^2,
\end{aligned}
\tag{119}
$$

where (a) follows from (116) and (118) and (b) follows from (104). Note that

$$\frac{1}{n}\sum_{j=0}^{n-1}s_j^2 - (\frac{1}{n}\sum_{j=0}^{n-1}s_j)^2 = \frac{n-1}{n}s^2, \tag{120}$$

where

$$s^2 := \frac{1}{n-1}\sum_{j=0}^{n-1}(s_j - \bar{s})^2, \tag{121}$$

is the famous statistic called "unbiased sample variance", and its variance is (see Page 229, Theorem 2 in [35])

$$\text{var}[s^2] = \frac{1}{n}\left(\mu_4 - \frac{n-3}{n-1}\mu_2^2\right), \tag{122}$$

where

$$\mu_4 = \mathbb{E}[(s_0 - \mu)^4], \tag{123}$$

and

$$\mu_2 = \mathbb{E}[(s_0 - \mu)^2] = \text{var}[s_0] = v. \tag{124}$$

Also note that the sample variance is unbiased, which means

$$\mathbb{E}_f[s^2] = v. \tag{125}$$

Therefore, we have

$$\mathbb{E}_f[(s^2)^2] = \text{var}[s^2] + (\mathbb{E}_f[s^2])^2 = \frac{1}{n}\left(\mu_4 - \frac{n-3}{n-1}v^2\right) + v^2, \tag{126}$$

so we have

$$
\begin{aligned}
\text{var}_f&\left[\frac{1}{n-k}\text{trace}(\mathbf{J}_2\mathbf{J}_2^\top)\right] \\
&\overset{(a)}{\leq} \mathbb{E}_f\left[\left(\frac{1}{n}\sum_{j=0}^{n-1}s_j^2 - (\frac{1}{n}\sum_{j=0}^{n-1}s_j)^2\right)^2\right] - \frac{k^2}{n^2}v^2 \\
&\overset{(b)}{=} \mathbb{E}_f\left[(\frac{n-1}{n}s^2)^2\right] - \frac{k^2}{n^2}v^2 \\
&= \frac{(n-1)^2}{n^2}\mathbb{E}_f[(s^2)^2] - \frac{k^2}{n^2}v^2 \\
&\overset{(c)}{=} \frac{(n-1)^2}{n^2}\frac{1}{n}\left(\mu_4 - \frac{n-3}{n-1}v^2\right) + \frac{(n-1)^2 - k^2}{n^2}v^2 \\
&= \mathcal{O}\left(\frac{1}{n}\right) + \frac{(n-1)^2 - k^2}{n^2}v^2,
\end{aligned}
\tag{127}
$$

where (a) follows from (119), (b) follows from (120) and (c) follow from (126).

Note that we have computed the expectation of $\frac{1}{n-k}\text{trace}(\mathbf{J}_2\mathbf{J}_2^\top)$, which is $\frac{k}{n}v$ (see (116)). Using the Chebyshev's inequality

$$
\Pr\left(\left|\frac{1}{n-k}\text{trace}(\mathbf{J}_2\mathbf{J}_2^\top) - \frac{k}{n}v\right| \geq \epsilon\right) \leq \frac{1}{\epsilon^2}\text{var}\left[\frac{1}{n-k}\text{trace}(\mathbf{J}_2\mathbf{J}_2^\top)\right]
$$

$$
\overset{(a)}{\leq} \frac{1}{\epsilon^2}\mathcal{O}\left(\frac{1}{n}\right) + \frac{1}{\epsilon^2}\frac{(n-1)^2 - k^2}{n^2}v^2
$$

$$
= \frac{1}{\epsilon^2}\mathcal{O}\left(\frac{1}{n}\right) + \frac{1}{\epsilon^2}\frac{(n-k-1)(n+k-1)}{n^2}v^2 \qquad (128)
$$

$$
\overset{(b)}{<} \frac{1}{\epsilon^2}\mathcal{O}\left(\frac{1}{n}\right) + \frac{2}{\epsilon^2}\frac{n-k-1}{n}v^2
$$

$$
= \frac{1}{\epsilon^2}\mathcal{O}\left(\frac{n-k}{n}\right).
$$

where (a) is from (127) and (b) is because $n + k - 1 < 2n$. Therefore, the proof of (114) is over. $\qquad\square$

Next, we show that with high probability the largest eigenvalue of $(\mathbf{J}_4)_{(n-k)\times(n-k)}$ is smaller than $(1+\epsilon)\mu$. Note that the matrix $\mathbf{J}_4$ is a principle submatrix of the Toeplitz matrix $\mathbf{F}\mathbf{\Lambda}\mathbf{F}^\top = \text{Toeplitz}[\tilde{s}_p]_{p=0,1,\dots,n-1}$, so $\mathbf{J}_4 = \text{Toeplitz}[\tilde{s}_p]_{p=0,1,\dots,n-k-1}$ is also Toeplitz. Using the Gershgorin circle theorem, all eigenvalues of $\mathbf{J}_4 := [\tilde{\mathbf{J}}_{ij}]$ must lie in the union of $(n-k)$ circles, in which the $i$-th circle is centered at the diagonal entry $\tilde{\mathbf{J}}_{ii} = \tilde{s}_0$ and has radius $\sum_{j\neq i}|\tilde{\mathbf{J}}_{ij}| = \sum_{j\neq i}|\tilde{s}_{j-i}|$. These $(n-k)$ circles are all within the circle centered at $\tilde{s}_0$ with radius $2\sum_{p=1}^{n-k-1}|\tilde{s}_p|$. Therefore, the maximum eigenvalue of $\mathbf{J}_4$ satisfies

$$
\sigma_{\max} < \tilde{s}_0 + 2\sum_{p=1}^{n-k-1}|\tilde{s}_p|. \qquad (129)
$$

Thus,

$$
\Pr(\sigma_{\max} > \mu + \epsilon) < \Pr\left(\tilde{s}_0 + 2\sum_{p=1}^{n-k-1}|\tilde{s}_p| > \mu + \epsilon\right)
$$

$$
= \Pr\left(\left(\tilde{s}_0 - \mu + 2\sum_{p=1}^{n-k-1}|\tilde{s}_p|\right)^2 > \epsilon^2\right)
$$

$$
\overset{(a)}{\leq} \frac{1}{\epsilon^2}\mathbb{E}\left[\left(\tilde{s}_0 - \mu + 2\sum_{p=1}^{n-k-1}|\tilde{s}_p|\right)^2\right] \qquad (130)
$$

$$
\overset{(b)}{\leq} \frac{1}{\epsilon^2}(2n - 2k - 1)^2\frac{v}{n} = \frac{1}{\epsilon^2}\mathcal{O}\left(\frac{(n-k)^2}{n}\right),
$$

where (a) is from the Markov inequality and (b) is due to the fact that $\text{var}[\tilde{s}_p] = \frac{v}{n}$ for all $p$ and $\mathbb{E}[\tilde{s}_0] = \mu$ and $\mathbb{E}[\tilde{s}_p] = 0$ for all $p \neq 0$.

From Lemma 8.4 and (130), when $n \to \infty$ and $(n-k)^2 = o(n)$, with high probability (which is $1 - \frac{1}{\epsilon^2}\mathcal{O}\left(\frac{(n-k)^2}{n}\right)$),

$$
\frac{1}{n-k}\text{trace}(\mathbf{J}_2\mathbf{J}_2^\top) \geq \frac{k}{n}v - \epsilon, \qquad (131)
$$

and at the same time

$$
\mathbf{J}_4^{-1} \succeq \frac{1}{\mu + \epsilon}\mathbf{I}_{n-k}. \qquad (132)
$$

From concentration of $\text{trace}(\mathbf{J}_2\mathbf{J}_2^\top)$ and the lower bound of $\mathbf{J}_4^{-1}$, we have, with high probability,

$$
\frac{1}{n-k}\text{trace}(\mathbf{J}_2\mathbf{J}_4^{-1}\mathbf{J}_2^\top) \geq \frac{1}{\mu + \epsilon}\left(\frac{k}{n}v - \epsilon\right), \qquad (133)
$$

for all $\epsilon$. This concludes the proof of Lemma 8.3 and hence completes the proof of Theorem 4.4 (see the details from after Lemma 8.3 to equation (113)). This lemma is a formal statement of equality (110).

## 8.8 Proof of Theorem 4.5

### 1) Uncoded linear inverse problem:

Consider the eigenvalue decomposition

$$\mathbf{B} = \mathbf{P}\Theta\mathbf{P}^{-1}, \tag{134}$$

where

$$\Theta = \text{diag}\{\gamma_1, \gamma_2, \ldots \gamma_N\}, \tag{135}$$

and without the loss of generality, assume $\gamma_1$ is the maximum eigenvalue. Then, from the definition $\mathbf{C}(l_i) = \mathbf{B}^{l_i}\mathbf{C}_E(\mathbf{B}^\top)^{l_i}$ in (9),

$$\mathbf{C}(l_i) = \mathbf{P}\Theta^{l_i}\mathbf{P}^{-1}\mathbf{C}_E(\mathbf{P}^\top)^{-1}\Theta^{l_i}\mathbf{P}^\top. \tag{136}$$

Since $\mathbf{P}^{-1}\mathbf{C}_E(\mathbf{P}^\top)^{-1}$ is a positive definite matrix, all of its eigenvalues are positive real numbers. Suppose the maximum eigenvalue and the minimum eigenvalue of $\mathbf{P}^{-1}\mathbf{C}_E(\mathbf{P}^\top)^{-1}$ are respectively $e_{\max}$ and $e_{\min}$. Then, (136) gives the upper and lower bounds

$$\text{trace}(\mathbf{C}(l_i)) \leq e_{\max}\text{trace}(\mathbf{P}\Theta^{2l_i}\mathbf{P}^\top), \tag{137}$$

and

$$\text{trace}(\mathbf{C}(l_i)) \geq e_{\min}\text{trace}(\mathbf{P}\Theta^{2l_i}\mathbf{P}^\top). \tag{138}$$

Suppose the maximum and minimum eigenvalues of $\mathbf{P}^\top\mathbf{P}$ are respectively $c_{\max}$ and $c_{\min}$. Then, (137) and (138) can be further simplified to

$$
\begin{aligned}
\text{trace}(\mathbf{C}(l_i)) &\leq e_{\max}\text{trace}(\Theta^{l_i}\mathbf{P}^\top\mathbf{P}\Theta^{l_i}) \\
&\leq c_{\max}e_{\max}\text{trace}(\Theta^{2l_i}) \\
&= c_{\max}e_{\max}\sum_{j=1}^{N}\gamma_j^{2l_i},
\end{aligned} \tag{139}
$$

and

$$
\begin{aligned}
\text{trace}(\mathbf{C}(l_i)) &\geq e_{\min}\text{trace}(\Theta^{l_i}\mathbf{P}^\top\mathbf{P}\Theta^{l_i}) \\
&\geq c_{\min}e_{\min}\text{trace}(\Theta^{2l_i}) \\
&= c_{\min}e_{\min}\sum_{j=1}^{N}\gamma_j^{2l_i},
\end{aligned} \tag{140}
$$

where the last equality in the above two inequalities are from the definition of $\Theta$ in (135). Therefore,

$$
\begin{aligned}
\lim_{T_{\text{dl}}\to\infty}\frac{1}{T_{\text{dl}}}\log\mathbb{E}[\|\mathbf{E}_{\text{uncoded}}\|^2\,|\mathbf{l}] &\overset{(a)}{=} \lim_{T_{\text{dl}}\to\infty}\frac{1}{T_{\text{dl}}}\log\left(\sum_{i=1}^{k}\text{trace}\left(\mathbf{C}(l_i)\right)\right) \\
&\overset{(b)}{=} \lim_{T_{\text{dl}}\to\infty}\frac{1}{T_{\text{dl}}}\log\left(\sum_{j=1}^{N}\sum_{i=1}^{k}\gamma_j^{2l_i}\right) \\
&= \lim_{T_{\text{dl}}\to\infty}\frac{1}{T_{\text{dl}}}\log\left(\sum_{j=1}^{N}\sum_{i=1}^{k}\gamma_j^{2\lceil\frac{T_{\text{dl}}}{v_i}\rceil}\right) \\
&\overset{(c)}{=} \max_{i\in[k],j}\log(\gamma_j)^{\frac{2}{v_i}},
\end{aligned} \tag{141}
$$

where (a) is from Theorem 4.1, (b) is obtained by plugging in (139) and (140) and the fact that the constants $e_{\min}$, $c_{\min}$, $e_{\max}$ and $c_{\min}$ do not change the error exponent when $T_{dl}$ increases, and (c) is

from the fact that the maximum term dominates the error exponent in a log-sum form. Since the maximum eigenvalue of the matrix $\mathbf{B}$ is $\gamma_1$, we have

$$
\begin{aligned}
\lim_{T_{\text{dl}} \to \infty} \frac{1}{T_{\text{dl}}} \log \mathbb{E}[\|\mathbf{E}_{\text{uncoded}}\|^2 \,|\mathbf{l}] &= \max_{i \in [k]} \log \gamma_1^{\frac{2}{v_i}} \\
&= -\frac{2}{\max_{i \in [k]} v_i} \log \frac{1}{\gamma_1}.
\end{aligned}
\tag{142}
$$

Therefore, the error exponent is determined by the worker with the slowest speed (maximum $v_i$).

**2) replication-based linear inverse:**

Now we look at the replication-based linear inverse scheme. At first, we do not know the order of the random sequence $v_1, v_2, \ldots v_n$. Therefore, when we assign the extra $n - k < k$ workers to replicate the computations of the last $n - k$ linear inverse problems, there is a non-zero probability that the slowest worker of the first $k$ workers does not have any other copy. More precisely, denote by $E$ the above event. Then, if we uniformly choose $n - k$ workers to replicate, the probability of $E$ is

$$
\Pr(E) = \frac{\binom{k-1}{n-k}}{\binom{k}{n-k}}.
\tag{143}
$$

This is also a constant that does not depend on the time $T_{\text{dl}}$. Therefore,

$$
\mathbb{E}[\|\mathbf{E}_{\text{rep}}\|^2 \,|\mathbf{l}] \geq c_{\min} e_{\min} \Pr(E) \sum_{j=1}^{N} \gamma_j^{2\left\lceil \frac{T_{\text{dl}}}{\max_{i \in [k]} v_i} \right\rceil},
\tag{144}
$$

where the exponent $2\left\lceil \frac{T_{\text{dl}}}{\max_{i \in [k]} v_i} \right\rceil$ is because we are lower-bounding the error of replication-based scheme using only the error of the slowest worker in the first $k$ workers, and $\Pr(E)$ is the probability that this particular worker is not replicated using any of the $n - k$ extra workers.

Using the fact that $\max_j \gamma_j = \gamma_1$ and the fact that $c_{\min} e_{\min} \Pr(E)$ is a constant that does not change with $T_{dl}$, we have

$$
\lim_{T_{\text{dl}} \to \infty} \frac{1}{T_{\text{dl}}} \log \mathbb{E}[\|\mathbf{E}_{\text{rep}}\|^2 \,|\mathbf{l}] \geq \frac{2}{\max_{i \in [k]} v_i} \log \frac{1}{\gamma_1}.
\tag{145}
$$

Note that $\mathbb{E}[\|\mathbf{E}_{\text{rep}}\|^2 \,|\mathbf{l}] \leq \mathbb{E}[\|\mathbf{E}_{\text{uncoded}}\|^2 \,|\mathbf{l}]$, so we also have

$$
\lim_{T_{\text{dl}} \to \infty} \frac{1}{T_{\text{dl}}} \log \mathbb{E}[\|\mathbf{E}_{\text{rep}}\|^2 \,|\mathbf{l}] \leq \lim_{T_{\text{dl}} \to \infty} \frac{1}{T_{\text{dl}}} \log \mathbb{E}[\|\mathbf{E}_{\text{uncoded}}\|^2 \,|\mathbf{l}] = \frac{2}{\max_{i \in [k]} v_i} \log \frac{1}{\gamma_1}.
\tag{146}
$$

Therefore,

$$
\lim_{T_{\text{dl}} \to \infty} \frac{1}{T_{\text{dl}}} \log \mathbb{E}[\|\mathbf{E}_{\text{rep}}\|^2 \,|\mathbf{l}] = \frac{2}{\max_{i \in [k]} v_i} \log \frac{1}{\gamma_1}.
\tag{147}
$$

**3) Coded linear inverse algorithm:**

For the coded linear inverse algorithm,

$$
\mathbb{E}[\|\mathbf{E}_{\text{coded}}\|^2 \,|\mathbf{l}] \leq \sigma_{\max}(\mathbf{G}^\top \mathbf{G}) \text{trace}\left[(\mathbf{G}\mathbf{\Lambda}^{-1}\mathbf{G}^\top)^{-1}\right].
\tag{148}
$$

From (139), we have

$$
\text{trace}(\mathbf{C}(l_i)) \leq c_{\max} e_{\max} \sum_{j=1}^{N} \gamma_j^{2l_i} \leq c_{\max} e_{\max} N \gamma_1^{2l_i}.
\tag{149}
$$

Plugging into (148), we have

$$
\mathbb{E}[\|\mathbf{E}_{\text{coded}}\|^2 \,|\mathbf{l}] \leq \sigma_{\max}(\mathbf{G}^\top \mathbf{G}) \text{trace}\left[(\mathbf{G}\mathbf{\Lambda}_2^{-1}\mathbf{G}^\top)^{-1}\right],
\tag{150}
$$

where

$$
\begin{aligned}
\mathbf{\Lambda}_2 &:= \text{diag}\{c_{\max} e_{\max} N \gamma_1^{2l_1}, \ldots c_{\max} e_{\max} N \gamma_1^{2l_n}\} \\
&= N c_{\max} e_{\max} \text{diag}\{\gamma_1^{2l_1}, \ldots \gamma_1^{2l_n}\}.
\end{aligned}
\tag{151}
$$

Since $N$, $c_{\max}$ and $e_{\max}$ are all constant numbers,

$$\lim_{T_{\mathrm{dl}}\to\infty}\frac{1}{T_{\mathrm{dl}}}\log\mathbb{E}[\|\mathbf{E}_{\mathrm{coded}}\|^2\,|\mathbf{l}]\leq\lim_{T_{\mathrm{dl}}\to\infty}\frac{1}{T_{\mathrm{dl}}}\log\mathrm{trace}\left[(\mathbf{G}\boldsymbol{\Lambda}_3^{-1}\mathbf{G}^\top)^{-1}\right],\tag{152}$$

where

$$\boldsymbol{\Lambda}_3:=\mathrm{diag}\{\gamma_1^{2l_1},\ldots\gamma_1^{2l_n}\}=\mathrm{diag}\{\gamma_1^{2\lceil\frac{T_{\mathrm{dl}}}{v_1}\rceil},\ldots\gamma_1^{2\lceil\frac{T_{\mathrm{dl}}}{v_n}\rceil}\}.\tag{153}$$

Define $\mathcal{S}=\{i_1,\ldots i_k\}$, i.e., the index set of the fastest $k$ workers. Then,

$$\min_{i\in\mathcal{S}}\left(\frac{1}{\gamma_1}\right)^{2\lceil\frac{T_{\mathrm{dl}}}{v_i}\rceil}=\left(\frac{1}{\gamma_1}\right)^{2\lceil\frac{T_{\mathrm{dl}}}{v_{i_k}}\rceil}.\tag{154}$$

For $i\in[n]\setminus\mathcal{S}=\{i_{k+1},\ldots i_n\}$,

$$\left(\frac{1}{\gamma_1}\right)^{2\lceil\frac{T_{\mathrm{dl}}}{v_i}\rceil}\geq 0.\tag{155}$$

Therefore, from the definition of the diagonal matrix $\boldsymbol{\Lambda}_3$ in (153), the entries of $\boldsymbol{\Lambda}_3^{-1}$ can be lower-bounded by (154) for $i\in\mathcal{S}$, and can be lower-bounded by (155) for $i\in[n]\setminus\mathcal{S}$. Thus,

$$\boldsymbol{\Lambda}_3^{-1}\succeq\left(\frac{1}{\gamma_1}\right)^{2\lceil\frac{T_{\mathrm{dl}}}{v_{i_k}}\rceil}\mathrm{diag}\{c_1,c_2,\ldots c_n\},\tag{156}$$

where $c_i$ is the indicator

$$c_i=\delta(i\in\mathcal{S}).\tag{157}$$

Define $\mathbf{G}_{\mathcal{T}}$ as the submatrix of $\mathbf{G}$ composed of the columns in $\mathbf{G}$ with indexes in $\mathcal{T}\subset[n]$. Use $\sigma_{\min}(\mathbf{X})$ to denote the minimum eigenvalue of a matrix $\mathbf{X}$. Define

$$s_{\min}=\min_{\mathcal{T}\subset[n],|\mathcal{T}|=k}\sigma_{\min}(\mathbf{G}_{\mathcal{T}}\mathbf{G}_{\mathcal{T}}^\top).\tag{158}$$

Since $\mathbf{G}$ is a matrix with orthonormal rows, any arbitrary $\mathbf{G}_{\mathcal{T}}$ that satisfies $|\mathcal{T}|=k$ must have full rank. This means that $s_{\min}>0$. Note that $s_{\min}>0$ is a constant that depends only on the generator matrix $\mathbf{G}$ and does not change with the overall time $T_{\mathrm{dl}}$. Therefore,

$$\begin{aligned}\mathbf{G}\boldsymbol{\Lambda}_3^{-1}\mathbf{G}^\top &\overset{(a)}{\succeq}\left(\frac{1}{\gamma_1}\right)^{2\lceil\frac{T_{\mathrm{dl}}}{v_{i_k}}\rceil}\mathbf{G}\mathrm{diag}\{c_1,c_2,\ldots c_n\}\mathbf{G}^\top\\[2mm]&\overset{(b)}{=}\left(\frac{1}{\gamma_1}\right)^{2\lceil\frac{T_{\mathrm{dl}}}{v_{i_k}}\rceil}\mathbf{G}_{\mathcal{S}}\mathbf{G}_{\mathcal{S}}^\top\\[2mm]&\overset{(c)}{\succeq}\left(\frac{1}{\gamma_1}\right)^{2\lceil\frac{T_{\mathrm{dl}}}{v_{i_k}}\rceil}s_{\min}\mathbf{I}_k.\end{aligned}\tag{159}$$

where (a) is from (156), (b) is from (157), and (c) is from (158). Thus, plugging (159) into (152) (note that there is an inverse inside the trace of (152))

$$\begin{aligned}\lim_{T_{\mathrm{dl}}\to\infty}\frac{1}{T_{\mathrm{dl}}}\log\mathbb{E}[\|\mathbf{E}_{\mathrm{coded}}\|^2\,|\mathbf{l}]&\leq\lim_{T_{\mathrm{dl}}\to\infty}\frac{1}{T_{\mathrm{dl}}}\log\mathrm{trace}\left[(\mathbf{G}\boldsymbol{\Lambda}_3^{-1}\mathbf{G}^\top)^{-1}\right]\\[2mm]&\leq\lim_{T_{\mathrm{dl}}\to\infty}\frac{1}{T_{\mathrm{dl}}}\log\mathrm{trace}\left[\left(\frac{1}{\gamma_1}\right)^{-2\lceil\frac{T_{\mathrm{dl}}}{v_{i_k}}\rceil}\frac{1}{s_{\min}}\mathbf{I}_k\right]\\[2mm]&=\lim_{T_{\mathrm{dl}}\to\infty}\frac{1}{T_{\mathrm{dl}}}\log\left\{\left(\frac{1}{\gamma_1}\right)^{-2\lceil\frac{T_{\mathrm{dl}}}{v_{i_k}}\rceil}\mathrm{trace}\left[\frac{1}{s_{\min}}\mathbf{I}_k\right]\right\}\\[2mm]&\overset{(a)}{=}\lim_{T_{\mathrm{dl}}\to\infty}\frac{1}{T_{\mathrm{dl}}}\log\left(\frac{1}{\gamma_1}\right)^{-2\lceil\frac{T_{\mathrm{dl}}}{v_{i_k}}\rceil}\\[2mm]&=-\frac{2}{v_{i_k}}\log\frac{1}{\gamma_1},\end{aligned}\tag{160}$$

where (a) is because $\mathrm{trace}\left[\frac{1}{s_{\min}}\mathbf{I}_k\right]=\frac{k}{s_{\min}}$ is a constant and does not change the error exponent. Thus, we have completed the proof of Theorem 4.5.

## 8.9 Computing the Matrix $\Lambda$

One difficulty in our coded linear inverse algorithm is pre-computing the entries $\text{trace}(\mathbf{C}(l)) = \text{trace}\left(\mathbf{B}^l \mathbf{C}_E (\mathbf{B}^\top)^l\right)$ in the weight matrix $\Lambda$ in (8), which involves a number of matrix-matrix multiplications. One way to side-step this problem is to estimate $\text{trace}(\mathbf{C}(l))$ using Monte Carlo simulations. Concretely, choose $m$ i.i.d. $N$-variate random vectors $\mathbf{a}_1, \mathbf{a}_2, \ldots \mathbf{a}_m$ that are distributed the same as the initial error $\mathbf{e}^{(0)}$ after Assumption 1. Then, compute the statistic

$$\hat{\gamma}_{m,l} = \frac{1}{m} \sum_{j=1}^{m} \left\| \mathbf{B}^l \mathbf{a}_j \right\|^2, l = 1, 2, \ldots T_u, \tag{161}$$

where $T_u$ is an upper bound of the number of iterations in a practical iterative computing algorithm. The Lemma 8.5 below shows that $\hat{\gamma}_{m,l}$ is an unbiased and asymptotically consistent estimator of $\text{trace}(\mathbf{C}(l))$ for all $l$. The computational complexity of computing $\hat{\gamma}_{m,l}, l = 1, 2, \ldots T_u$ is the same as the computation of $m$ linear inverse problems for $T_u$ iterations. The computation has low complexity and can be carried out distributedly in $m$ workers before the main algorithm starts. Additionally, the computation results can be used repeatedly when we implement the coded linear inverse algorithm multiple times. In our experiments on PageRank, for each graph we choose $m = 10$ and estimate $\text{trace}(\mathbf{C}(l))$ before implementing the coded linear inverse algorithm (in this case it is the coded power-iteration algorithm), which has the same complexity as solving $m = 10$ extra linear inverse problems.

*Lemma 8.5.* The statistic $\hat{\gamma}_{m,l}$ is an unbiased and asymptotically consistent estimator of $\text{trace}(\mathbf{C}(l))$. More specifically, the mean and variance of the estimator $\hat{\gamma}_{m,l}$ satisfies

$$\mathbb{E}[\hat{\gamma}_{m,l}|\mathbf{l}] = \text{trace}(\mathbf{C}(l)), \tag{162}$$

$$\text{var}_t[\hat{\gamma}_{m,l}] \leq \frac{1}{m} \left\| \mathbf{B}^l \right\|_F^4 \mathbb{E}\left[\left\| \mathbf{a}_j \right\|^4\right]. \tag{163}$$

*Proof.* The expectation of $\hat{\gamma}_{m,l}$ satisfies

$$
\begin{aligned}
\mathbb{E}[\hat{\gamma}_{m,l}] &= \frac{1}{m} \sum_{j=1}^{m} \mathbb{E}\left[\left\| \mathbf{B}^l \mathbf{a}_j \right\|^2\right] \\
&= \mathbb{E}\left[\left\| \mathbf{B}^l \mathbf{a}_1 \right\|^2\right] \\
&= \mathbb{E}\left[\text{trace}(\mathbf{B}^l \mathbf{a}_1 \mathbf{a}_1^\top (\mathbf{B}^l)^\top)\right] \\
&\stackrel{(a)}{=} \text{trace}(\mathbf{B}^l \mathbb{E}[\mathbf{a}_1 \mathbf{a}_1^\top](\mathbf{B}^l)^\top) \\
&= \text{trace}(\mathbf{B}^l \mathbf{C}_E (\mathbf{B}^l)^\top) \\
&= \text{trace}(\mathbf{C}(l)),
\end{aligned} \tag{164}
$$

where (a) is from the fact that $\mathbf{a}_1$ has covariance $\mathbf{C}_E$. To bound the variance of $\hat{\gamma}_{m,l}$, note that for all $j$,

$$\left\| \mathbf{B}^l \mathbf{a}_j \right\|^2 \leq \left\| \mathbf{B}^l \right\|_F^2 \left\| \mathbf{a}_j \right\|^2. \tag{165}$$

Therefore,

$$
\begin{aligned}
\text{var}[\hat{\gamma}_{m,l}] &= \text{var}[\frac{1}{m} \sum_{j=1}^{m} \left\| \mathbf{B}^l \mathbf{a}_j \right\|^2] \\
&\stackrel{(a)}{=} \frac{1}{m} \text{var}\left[\left\| \mathbf{B}^l \mathbf{a}_j \right\|^2\right] \\
&\stackrel{(b)}{\leq} \frac{1}{m} \mathbb{E}\left[\left\| \mathbf{B}^l \mathbf{a}_j \right\|^4\right] \\
&\stackrel{(c)}{\leq} \frac{1}{m} \left\| \mathbf{B}^l \right\|_F^4 \mathbb{E}\left[\left\| \mathbf{a}_j \right\|^4\right],
\end{aligned} \tag{166}
$$

where (a) holds because all $\|\mathbf{a}_j\|$ are independent of each other, and (b) holds because $\text{var}[X] \leq \mathbb{E}[X^2]$, and (c) is from the Cauchy-Schwartz inequality. $\qquad \square$

For the correlated case, we have to compute a slightly modified weighting matrix denoted by $\tilde{\boldsymbol{\Lambda}}$ in (34). The only change is that we have to compute $\Psi_{i,j}$ in (35) for all possible $l_i, l_j$ such that $1 \leq l_i, l_j \leq T_u$. We also choose $m$ i.i.d. $N$-variate random vectors $\mathbf{b}_1, \mathbf{b}_2, \ldots \mathbf{b}_m$ that are distributed with mean $\mathbf{0}_N$ and covariance $\mathbf{C}_{\text{cor}}$, which is the same as the correlation part according to Assumption 4. Then, compute the statistic

$$\hat{\gamma}_{m,(l_i,l_j)} = \frac{1}{m} \sum_{u=1}^{m} \mathbf{b}_u \mathbf{B}^{l_j} \mathbf{B}^{l_i} \mathbf{b}_u, 1 \leq l_i, l_j \leq T_u. \tag{167}$$

Then, it is easy to show that $\hat{\gamma}_{m,(l_i,l_j)}$ is also an unbiased and asymptotically consistent estimator of $\Psi_{i,j}$.

## 8.10 Proof of Theorem 5.1

The computational complexity at each worker is equal to the number of operations in one iteration multiplied by the number of iterations. The number of iterations is $l$. In each iteration, the number of operations is equal to the number of non-zeros in $\mathbf{B}$ because each iteration $\mathbf{x}^{(l+1)} = \mathbf{Kr} + \mathbf{Bx}^{(l)}$ requires at least scanning through the non-zeros in $\mathbf{B}$ once to compute $\mathbf{Bx}^{(l)}$. Note that we only count the number of non-zeros in $\mathbf{B}$ and ignore the matrix $\mathbf{K}$ because $\mathbf{K}$ may not be a square matrix. In fact, it can even be a scalar in the PageRank problem[2]. For general $\mathbf{B}$ matrices, the number of entries is in the order of $N^2$, where $N$ is the number of rows in $\mathbf{B}$. Therefore, the overall number of operations at each worker is in the order of $\Theta(N^2 l)$.

The encoding and decoding steps in Algorithm 1 are all based on matrix-matrix multiplications. More specifically, for encoding, we multiply the generator matrix $\mathbf{G}_{k \times n}$ with the input matrix and the initial estimates, which both have size $N \times k$. Thus, the complexity scales as $\mathcal{O}(nkN)$. For decoding, the computation of the decoding matrix $\mathbf{L} = (\mathbf{G}^\top \boldsymbol{\Lambda}^{-1} \mathbf{G})^{-1} \mathbf{G} \boldsymbol{\Lambda}^{-1}$ is has complexity $\Theta(k^3)$ (matrix inverse) plus $\Theta(k^2 n)$ (matrix multiplications). Multiplying the decoding matrix $\mathbf{L}_{k \times n}$ with linear inverse results that have size $N \times n$ has complexity $\Theta(nkN)$. Therefore, for large $N$, the computational complexity is in the order of $\Theta(nkN)$.

The computation of the matrix $\boldsymbol{\Lambda}$, as we have explained in Section 8.9, has the same complexity as computing $m \approx 10$ extra linear inverse problems. Additionally, it is a one-time cost in the pre-processing step. Thus, we do not take into account the complexity of computing $\boldsymbol{\Lambda}$ for the analysis of encoding and decoding.

## 8.11 Proof of Theorem 5.2

We assume that the matrix $\mathbf{B}$ and $\mathbf{K}$ have already been stored in each worker before the computation of the linear inverse problems. For the PageRank problem, this means that we store the column-normalized adjacency matrix $\mathbf{A}$ in each worker.

In Algorithm 1, the $i$-th worker requires the central controller to communicate a vector $\mathbf{r}_i$ with length $N$ to compute the linear inverse problem. Thus

$$\text{COST}_{\text{communication}} = N \quad \text{INTEGERS.} \tag{168}$$

The computation cost at each worker is equal to the number of operations in one iteration multiplied by the number of iterations in the specified iterative algorithm. In each iteration, the number of operations also roughly equals to the number of non-zeros in $\mathbf{B}$. Thus

$$\text{COST}_{\text{computation}} \approx 2 \cdot |\mathcal{E}| \cdot l_i \text{ OPERATIONS,} \tag{169}$$

where $l_i$ is the number of iterations completed at the $i$-th worker, $|\mathcal{E}|$ is the number of non-zeros in $\mathbf{B}$, and $2\cdot$ is because we count both addition and multiplication. From Fig. 7, the typical number of $l_i$ is about 50.

Thus, the ratio between computation and communication is

$$\text{COST}_{\text{computation}}/\text{COST}_{\text{communication}}$$
$$\approx l_i \bar{d} \text{ OPERATIONS/INTEGERS,} \tag{170}$$

Figure 4: This simulation result shows the mean squared error of the computation results for $k = 200$ different problems in the uncoded scheme.

Figure 5: This simulation result shows the mean squared error of the computation results for $k = 200$ different problems in the coded scheme.

where $\bar{d}$ is the average number of non-zeros in each row of the $\mathbf{B}$ matrix (because $N$ is the number of rows in $\mathbf{B}$). Since $l_i$ is about 50, we expect that the computation cost is much larger than communication.

## 8.12 Simulations

We also test the coded linear inverse algorithm for the personalized PageRank problem in a simulated setup with randomly generated graphs and worker response times. These simulations help us understand looseness in our theoretical bounding techniques. They can also test the performance of the coded Algorithm for different distributions. We simulate Algorithm 1 on a randomly generated Erdös-Rényi graph with $N = 500$ nodes and connection probability 0.1. The number of workers $n$ is set to be 240 and the number of PageRank vectors $k$ is set to be 200. We use the first $k = 200$ rows of a $240 \times 240$ DFT-matrix as the $\mathbf{G}$ matrix in the coded PageRank algorithm in Section 3. In Fig. 4 and Fig. 5, we show the simulation result on the mean squared error of all $k = 200$ PageRank vectors in both uncoded and coded PageRank, which are respectively shown in Fig. 4 and

Figure 6: This figure shows the mean squared error of uncoded, replication-based and coded PageRank algorithms.

Fig. 5. The x-axis represents the computation results for different PageRank problems and the y-axis represents the corresponding mean-squared error. It can be seen that in the uncoded PageRank, some of the PageRank vectors have much higher error than the remaining ones (the blue spikes in Fig. 4), because these are the PageRank vectors returned by the slow workers in the simulation. However, in coded PageRank, the peak-to-average ratio of mean squared error is much lower than in the uncoded PageRank. This means that using coding, we are able to mitigate the straggler effect and achieve more uniform performance across different PageRank computations. From a practical perspective, this means that we can provide fairness to different PageRank queries.

We compare the average mean-squared error of uncoded, replication-based and coded PageRank algorithms in Fig. 6. The first simulation compares these three algorithms when the processing time of one iteration of PageRank computation is exponentially distributed, and the second and third when the number of iterations is uniformly distributed in the range from 1 to 20 and Bernoulli distributed at two points 5 and 20 (which we call "delta" distribution). It can be seen that in all three different types of distributions, coded PageRank beats the other two algorithms.

### 8.13 Validating Assumption 3 using Experiments

Here we provide an experiment that validates Assumption 3 in Section 4.3, i.e., the computation time of one power-iteration at the same worker is a constant over time. In Fig. 7, we plot the number of power-iterations completed at different workers versus computation time. We can see that the computation speed is indeed constant, which means that Assumption 3 is valid[3]. Note that there is a non-zero time cost for loading the graph at each worker. This amount of time does not exist if the network graph is already loaded in the cache of distributed workers for online queries. Additionally, this amount of cost does not affect the conclusion of Theorem 4.5, because the loading time becomes negligible when the computation deadline $T_{\text{dl}} \to \infty$.

### 8.14 Proof of Lemma 8.1

Property 1 and property 2 can be directly examined from the definition. Property 3 is Theorem 3 in [36]. To prove property 4, we note that the eigenvalues of $\mathbf{A} \otimes \mathbf{B}$ equals to the pairwise products of the eigenvalues of $\mathbf{A}$ and the eigenvalues of $\mathbf{B}$ (from Theorem 6 in [36]). Therefore, since the

Figure 7: This figure shows the number of PageRank power-iterations completed at different workers in 30 seconds in the Google Plus experiment.

eigenvalues of $\mathbf{A}$ and the eigenvalues of $\mathbf{B}$ are all non-negative, the eigenvalues of $\mathbf{A} \otimes \mathbf{B}$ are also non-negative. Property 5 follows directly from property 4 because when $\mathbf{B} - \mathbf{A} \succeq \mathbf{0}$ and $\mathbf{C} \succeq \mathbf{0}$, $(\mathbf{B} - \mathbf{A}) \otimes \mathbf{C} \succeq \mathbf{0}$.

To prove property 6, we can repeatedly use property 3:

$$
\begin{aligned}
(\mathbf{A}_{m \times n} \otimes \mathbf{I}_p) \cdot (\mathbf{I}_n \otimes \mathbf{B}_{p \times q}) &= (\mathbf{A}_{m \times n} \cdot \mathbf{I}_n) \otimes (\mathbf{I}_p \cdot \mathbf{B}_{p \times q}) \\
&= (\mathbf{I}_m \cdot \mathbf{A}_{m \times n}) \otimes (\mathbf{B}_{p \times q} \cdot \mathbf{I}_q) \\
&= (\mathbf{I}_m \otimes \mathbf{B}_{p \times q}) \cdot (\mathbf{A}_{m \times n} \otimes \mathbf{I}_q).
\end{aligned}
\tag{171}
$$

To prove property 7, we first assume that

$$
\mathbf{L}_{k \times n} = \begin{bmatrix} L_{11} & L_{12} & \dots & L_{1n} \\ L_{21} & L_{22} & \dots & L_{2n} \\ \vdots & \vdots & \ddots & \vdots \\ L_{n1} & L_{n2} & \dots & L_{kn} \end{bmatrix}.
\tag{172}
$$

Then,

$$
\begin{aligned}
\operatorname{trace}\left[ (\mathbf{L} \otimes \mathbf{I}_N) \cdot \mathbf{A} \cdot (\mathbf{L} \otimes \mathbf{I}_N)^\top \right] &\overset{(a)}{=} \sum_{l=1}^{k} \operatorname{trace}\left[ \sum_{i=1}^{n} \sum_{j=1}^{n} L_{ki} \mathbf{A}_{ij} L_{kj} \right] \\
&= \sum_{l=1}^{k} \left[ \sum_{i=1}^{n} \sum_{j=1}^{n} L_{ki} \operatorname{trace}[\mathbf{A}_{ij}] L_{kj} \right] \\
&\overset{(b)}{=} \operatorname{trace}\left[ \mathbf{L} \cdot \begin{bmatrix} \operatorname{trace}[\mathbf{A}_{11}] & \dots & \operatorname{trace}[\mathbf{A}_{1n}] \\ \vdots & \ddots & \vdots \\ \operatorname{trace}[\mathbf{A}_{n1}] & \dots & \operatorname{trace}[\mathbf{A}_{nn}] \end{bmatrix} \cdot \mathbf{L}^\top \right],
\end{aligned}
\tag{173}
$$

where (a) and (b) hold both because the trace can be computed by examining the trace on the diagonal (or the diagonal blocks).

## Footnotes

[2]Since we are trying to prove Theorem 5.1 which states that the computational cost is higher than that of the communication cost, neglecting the computation of $\mathbf{K}$ does not affect the result.

[3]In this work, we assume that the statistics of speed distributions are unknown. However, from Fig. 7, it may seem that the speeds at different workers are quite predictable. In fact, each time when scheduling tasks to the pool of parallel workers, the central controller assign the tasks through virtual machines instead of actual physical addresses. Therefore, the machines assigned to the same task can be different, and this assignment is transparent to the end-users. Thus, the statistics of speed distributions are generally unobtainable.