[Reviews · NeurIPS 2017]

Reviewer 1



The paper considers a distributed computing scenario where a bunch of processors have different delays in computing. A job (here finding a solution of a system of linear equations) is distributed among the processors. After a certain deadline the output of the processors are combined and the result is produced. Previous literature mainly neglect the slower processors and work with the best k out of n processors' outputs. Usually an MDS code is used to distribute the task among the processors. I understand the main contribution is to include the stragglers computations and not treat them as mere erasures. I have few critical comments regarding the work. 1. Correctness of results: I think there are serious problems in the assumptions (see line 116-117). If x_i s are all iid, it is not necessary that r_i s are all iid as well. This only happens when M is square invertible matrix. For example, if a row of M is zero then the corresponding entry of r_i is going to be always zero. The main results all depend on this assumption, which makes me doubtful about the veracity of results. 2. Assumptions without justification: From the introduction it seems like the main result is contained in Theorem 4.4. This theorem is true under two very specific assumptions. One, the solutions of a systems of linear equations are all iid variables, and the number of iterations of each processor within a deadline are all iid random variables. These two seem like very stringent condition and I see no reason for these to be true in real systems. The authors also do not provide any motivation for these assumptions to be true either. Given this, I am unable to judge whether the theoretical results are of any value here. Another key condition for the results to be true is n-k = o(\sqrt(n). Again it was not explained why this should be the case even asymptotically. Also, I do not understand the implication of the random variable C(l) satisfying the ratio of variance and mean-square to be large. Why would this assumption make any sense? Overall lack of motivations for these assumptions seem concerning. 3. Unfair comparison: The replication scheme, that the paper compare their coding scheme with, has several problems. And I do not think that it is a good comparison. The assumption is that n -k is very small, i.e., o(\sqrt{n}). So computation of only o(\sqrt{n}) elements are computed twice. The rest of the processors are uncoded. This is a very specific coding scheme, and it is obvious that the Fourier matrix based scheme (which is an MDS code) will beat this scheme. In the introduction it was claimed that: “To compare with the existing coding technique in [6], ..” But the theoretical comparison to the method of [6] is missing. 4. Novelty: I do not think the presented scheme is new. It is the same MDS code based scheme that was proposed by Lee et al (in ref [6]). Overall the same encoding method has been used in both [6] and in this paper. The only difference is that the results of the stragglers are also used in this paper while decoding - so it is expected to get a better error rate. But in general, the claim of introduction, the prevention of ill-conditioning of MDS matrices in the erasure channel, remains unmitigated. It is just that a particular problem is considered in the paper. 5. Experiments: 9. The figures are not properly labeled. In the third plot of fig 2, ref [12] is mentioned, which should be [6] (I think). This is quite confusing. I am further skeptical about this plot. At certain value of the T_deadline, the method of [6], should result in absolute 0 error. But this is not observed in the plot. It is also not clear why in the extension of the coded method the mean square error would increase. 6. More minor comments: The authors say “iterative methods for linear inverse problem” as if it is a very standard problem: however it is not to me. A more understandable term would be `finding solutions of a linear system'. So for quite long time I could not figure out what the exact problem they are trying to solve. This create difficulty reading the entirety of the introduction/abstract. \rho has been used as spectral radius, and also in the heavy tail distributions. ============= After the rebuttal: I am able to clarify my confusion about independence of the random variables. So I do not have about the concern with correctness anymore. The point that I am still worried about is the unfair comparison with repetition codes! I do not think n-k =o(n) is the correct regime to use repetition code. It seems to me from the rebuttal is, if T_deadline is large enough, then coding does not help (which makes sense). Is it clear what is the crossover point of T_deadline? I am satisfied with the rebuttal in other points.

Reviewer 2



The paper presents an error correcting based technique to speedup iterative linear solvers. The authors follow a new line of work that uses ideas from coding theory to alleviate the effect of straggler (eg slower than average) nodes in distributed computation, where in many works so far the computational result of straggler nodes is considered as erased information. The main idea of coding in the context of distributed computation is the following (a high-level, 3 node example): if a worker is computing task1, a second worker is computing task2, and a third is computing a “linear combination” of task1+task2, then from any 2 workers a master node could recover both computational tasks. This idea can both provably and in practice improve the performance of distributed algorithms for simple linear problems like matrix multiplication, or linear regression. In this work, the authors present an interesting new idea: for linear iterative solvers where we are optimizing with respect to some accuracy, instead of treating straggler nodes as erasures, we could still use some partial results from them. The algorithmic flow is as follow: the tasks are encoded to add redundancy, and the workers start their local computation. Then, the master node sets a deadline, after which it collects any results computed so far by the workers. Unlike, the “erasures” case, here partial results are used from all workers, and this non-discarded computational effort can provably help. The authors compare the coding technique with no coding and task replication, and establish that in theory when coding is used, then the accuracy of the distributed iterative linear solvers can be unboundedly better, than both replication or uncoded computation. The authors finally present experiments on HT-Condor with 120 worker nodes, where the coded scheme outperforms both the uncoded and replication schemes significantly. My only concern with the experiments is that the size of the data set to be parallelized is actually small enough to fit in a single machine, so it is unclear if any speedups can be observed in this case. It would be interesting to see how a larger data set would perform for these problems. Overall, the paper is well-written, it presents an interesting idea and analysis, and has novel and technically sound results. On the experimental side it would be interesting to see how things scale beyond medium sized data sets, and for the case where faster machines than Condor-HT (eg amazon ec2 gpu instances) are used.

Reviewer 3



The authors propose a new coded computation scheme for solving “many” linear inverse problems in parallel. While the existing works on coded computation mostly view the stragglers as ‘erasures’, the authors view the outputs of each processor, which are intermediate results of the iterative algorithms, as ‘noisy’ computation results. Under this new view, the authors propose a new encoding/decoding scheme, analyze the performance of their scheme, and compare it with other schemes based on the erasure view. The paper is concrete, and I believe that this paper brings a new (and useful) angle to the community. I have two minor concerns. One concern is that some parts of the paper are hard to follow, which I think make the paper hard to follow and grasp the key idea of it. Further, one key reference is missing: “Anytime Coding for Distributed Computation” by Ferdinand and Draper, Allerton 2016. To the best of my knowledge, although they are still under the umbrella of the erasure view, they are the first ones who study the performance of coded computation in terms of computing approximate computation results as a function of “stopping time”.